# Gogo: Group-wise granularity-ordered codec for stable and efficient speech generation

**Weidong Chen, Helen M. Meng, Xixin Wu**[*]
The Chinese University of Hong Kong
`{wdchen,hmmeng,wuxx}@se.cuhk.edu.hk`

## Abstract

Current speech language models require their core component, the speech codec, to discretize continuous speech signals into tokens that not only capture high-level cues for autoregressive modeling but also preserve sufficient acoustic details for perceptual quality. To address this need, we propose Gogo, a group-wise granularity-ordered codec that quantizes each group of frames into tokens arranged from coarse to fine, where coarse tokens encode high-level abstractions and fine tokens progressively recover low-level details. Building on the granularity-ordering property of Gogo, we introduce GogoSpeech, a two-stage speech language model that performs speech generation by first constructing a coarse speech backbone at an extremely low token rate and then enriching the backbone with fine-grained acoustic details. Considering the inherently non-uniform information distribution in speech signals, we further design a Group Relative Policy Optimization (GRPO)-trained token allocator that adaptively allocates token budgets to groups based on group-wise complexity. Experimental results demonstrate that Gogo delivers state-of-the-art reconstruction performance across most metrics at a token rate of 47. Moreover, evaluations on zero-shot text-to-speech tasks show that GogoSpeech enables efficient generation by adaptively reducing the average token rate, and attains state-of-the-art results in long-form speech generation.

## 1 Introduction

Large language models (LLMs) such as the GPT series (Brown et al., 2020; OpenAI, 2024) have demonstrated remarkable capabilities across diverse text-based tasks. Their success has inspired growing efforts to extend the LLM paradigm to the speech modality, leading to the development of speech language models (SLMs) capable of understanding and generating speech (Ye et al., 2025b; Défossez et al., 2024). A common pipeline for SLMs first discretizes speech into sequences of tokens via an audio codec, then models both text and speech tokens autoregressively. The effectiveness of this approach hinges critically on the codec's ability to produce tokens that concurrently contain high-level cues (*e.g.*, content, semantics, and structural attributes) for autoregressive modeling (Ye et al., 2025a) and low-level details (*e.g.*, acoustic fluctuations) for perceptual quality preservation.

Conventional audio codecs use frame-wise quantization for compression and transmission (Kleijn et al., 2021; Valin et al., 2012). While this enables high-fidelity reconstruction, its strong locality bias limits the codec's ability to capture high-level cues needed by SLMs. To address this limitation, recent works have augmented codecs with self-supervised representations (Zhang et al., 2024; Li et al., 2025) or automatic speech recognition (ASR) features (Jo et al., 2025; Zeng et al., 2025) to explicitly inject high-level linguistic and semantic information into the quantization process. However, the fundamental frame-wise quantization paradigm remains unbroken, inherently limiting the ability to learn high-level information. Besides, little attention has been given to the non-uniform information distribution in speech (Dieleman et al., 2021; Voran, 2024). Current approaches generally allocate one same bitrate to all segments, which leads to redundant coding and low generation efficiency, especially in less complex segments like silence where a low coding rate would suffice.

To address the aforementioned limitations, we redesign both the codec and SLM framework. As depicted in Figure 1, we first propose Gogo, a group-wise granularity-ordered codec that processes

---

[*]Corresponding author.

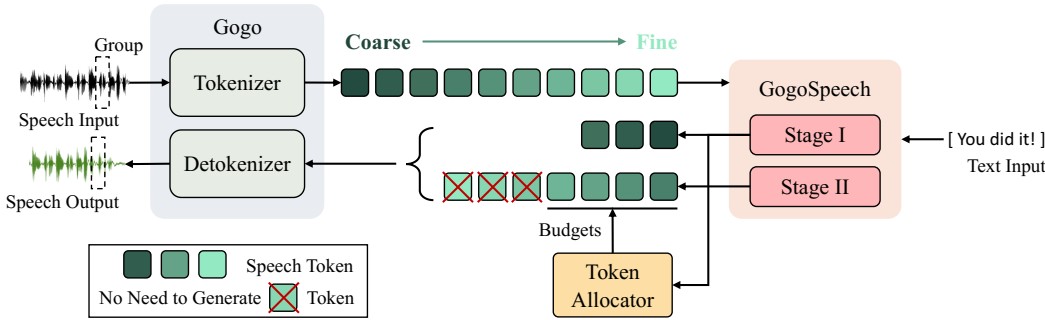

Figure 1: System overview. The shading of a token reflects the granularity of the information it encodes. The token allocator allocates different token budgets to different groups within the same utterance based on their complexity. Only one group is plotted for simplicity. Best viewed in color.

contiguous frames as groups and generates tokens in a coarse-to-fine order, where coarse tokens capture high-level information and fine tokens are used to progressively restore low-level acoustic details. Building on the granularity-ordering property of Gogo, we develop GogoSpeech, a two-stage SLM for speech generation. In the first stage, a high-level speech backbone, which serves as a coarse indicator of the target speech, is predicted at a extremely low feature rate (around 14 Hz). This reduced feature rate enhances the stability of autoregressive prediction and mitigates error accumulation (Arora et al., 2022; He et al., 2021). In the second stage, fine-grained details are incrementally recovered conditioned on the speech backbone. The feature rate in the second stage is restored to a standard level to ensure high-fidelity synthesis with precise details. To further enhance efficiency, we propose a Group Relative Policy Optimization (GRPO)-trained (Shao et al., 2024) token allocator that dynamically assigns token budgets to every group based on its complexity, thereby aligning computational resources with the non-uniform information density of speech signals.

Our main contributions in this work are fivefold: (1) proposing a new speech codec, namely Gogo, featuring group-by-group and granularity-ordered tokenization that better addresses the representational requirements of SLMs; (2) constructing a new speech language model, namely GogoSpeech, enabling staged speech generation from high-level abstractions to fine acoustic details; (3) developing a GRPO-trained token allocator that optimizes efficiency and quality by aligning token budgets with group-wise complexity; (4) conducting reconstruction experiments to show that Gogo achieves superior performance compared to state-of-the-art codecs; (5) performing experiments on text-to-speech (TTS) to show that GogoSpeech achieves better performance with higher stability and efficiency. Demo samples can be found at `https://happycolor.github.io/gogo`.

## 2 RELATED WORK

### 2.1 NEURAL AUDIO CODECS

Modern neural audio codecs are predominantly based on the VQ-GAN framework (Esser et al., 2021), which integrates an encoder, a vector quantizer, and a decoder into an end-to-end system. Pioneering works such as SoundStream (Zeghidour et al., 2021) and EnCodec (Défossez et al., 2022) employ residual vector quantization (RVQ) together with carefully designed discriminators to improve perceptual fidelity. DAC (Kumar et al., 2023) further enhances codebook utilization by performing code lookup in a low-dimensional space using cosine similarity rather than Euclidean distance. To improve compatibility with SLMs, recent studies have explored injecting linguistic or semantic information into the quantization process. One line of work leverages self-supervised speech representations from models such as HuBERT (Hsu et al., 2021), WavLM (Chen et al., 2022), and w2v-BERT (Chung et al., 2021). For example, SpeechTokenizer (Zhang et al., 2024) distills semantic teacher representations into the first stage of RVQ, while Mimi (Défossez et al., 2024) transfers semantic information into a single-stage quantizer to decouple acoustic reconstruction from semantic coding. Another line of research integrates ASR features into codec training. Notably, $\mathcal{S}^3$ tokenizer (Du et al., 2024a) partitions the encoder of a pretrained SenseVoice ASR model (An et al.,

2024) and inserts a quantization layer between its two halves. Despite recent advancements, the frame-wise quantization paradigm remains unchanged. Its inherent locality bias limits the codec's ability to learn high-level cues, which are essential for stable autoregressive modeling in SLMs.

## 2.2 SPEECH LANGUAGE MODELS

Generative speech language modeling extends the LLM paradigm to the speech modality by modeling discrete speech tokens produced by neural audio codecs or clustering methods such as k-means. Early works such as GSLM (Lakhotia et al., 2021) and SpeechGPT (Zhang et al., 2023) directly train language models on speech tokens, enabling the generation of natural-sounding speech. VALL-E (Wang et al., 2023) adopts an autoregressive model to generate the first token and a non-autoregressive model to predict residual tokens from EnCodec. To address the trade-off between high-fidelity reconstruction and effective autoregressive modeling, AudioLM (Borsos et al., 2023) introduces a hierarchical modeling framework that first generates semantic tokens and then refines them with acoustic tokens. In general, acoustic tokens are designed to encode speech at a low bitrate while preserving as much information as possible, whereas semantic tokens are learned from self-supervised speech models to capture phonetic or semantic representations that facilitate speech comprehension (Guo et al., 2025b). This hierarchical paradigm has inspired numerous follow-up works, including AudioPaLM (Rubenstein et al., 2023), Moshi (Défossez et al., 2024), and TTS-Llama (Shen et al., 2025). While the hierarchical approach improves stability and controllability, the semantic modeling stage in existing methods typically operates at the same token rate as the acoustic modeling stage, which is substantially higher than the token rate used in the text modality.

## 2.3 ADAPTIVE BITRATE IN NEURAL CODECS

The non-uniform information density of speech signals makes constant-bitrate codecs inherently inefficient. SNAC (Siuzdak et al., 2024) extends RVQ to operate at multiple temporal resolutions, yet the bitrate remains fixed across different speech regions. Acoustic BPE (Shen et al., 2024) applies the byte-pair encoding (BPE) algorithm (Devlin et al., 2019) to speech tokens, reducing sequence length and increasing token correlation. More recently, VRVQ (Chae et al., 2025) introduces a variable-bitrate strategy into RVQ, allowing the number of quantizers per frame to be adaptively determined from a predicted importance map. Similarly, TFC (Zhang et al., 2025) dynamically allocates frame rates to different regions according to temporal entropy. However, these variable-bitrate codecs do not explicitly couple bitrate variation with reconstruction quality for joint optimization, and their effectiveness in generative tasks within the SLM framework remains underexplored.

## 3 METHODS

### 3.1 GOGO

The proposed Gogo, as shown in Figure 2, comprises three main components: an encoder for learning speech representations, a flow-based generative model (Lipman et al., 2023; Tong et al., 2024) for mel-spectrogram reconstruction, and a vocoder for converting spectrograms into waveforms. Additionally, Gogo integrates an ASR module and an autoregressive (AR) prior (Wang et al., 2025; Yang et al., 2025) to enhance the suitability of the learned speech tokens for downstream generation.

### 3.1.1 WORKFLOW

Given an input waveform $w$, we first extract its mel-spectrogram $x \in \mathbb{R}^{n_f \times d}$, where $n_f$ denotes the number of frames and $d$ the number of mel bins. The spectrogram is then partitioned along the temporal axis into multiple non-overlap groups $x^i \in \mathbb{R}^{g \times d}$, where $g$ denotes the group size, $i \in [1, n_g]$ denotes the group index, and $n_g = \lceil \frac{n_f}{g} \rceil$ denotes the total number of groups. The last group is zero-padded if necessary. Subsequently, each group is concatenated with $n_q$ learnable queries $q^i \in \mathbb{R}^{n_q \times d}$, which yields $z^i \in \mathbb{R}^{(g+n_q) \times d}$. Incorporating queries into quantization has been explored in ALMTokenizer (Yang et al., 2025), TiTok (Yu et al., 2024), and FlexTok (Bachmann et al., 2025). The extended sequences $z^i$ are encoded by Transformer (Vaswani et al., 2017) encoder, after which the $x^i$ part is discarded and finite scalar quantization (FSQ) (Mentzer et al., 2024) is applied to the positions corresponding to the learnable queries, producing speech token indices $s^i \in$

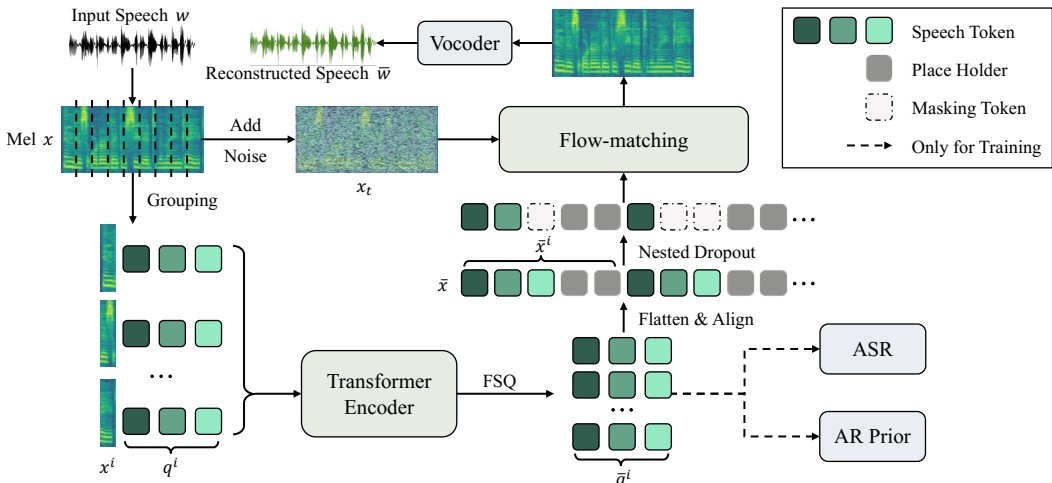

Figure 2: Architecture of Gogo. Here the group size $g$ is set to 5 and the number of speech queries $n_q$ assigned to each group is set to 3 for demonstration. Best viewed in color.

$\mathbb{R}^{n_q}$ and their corresponding embeddings $\bar{q}^i \in \mathbb{R}^{n_q \times d_h}$, where $d_h$ denotes the hidden dimension. Formally, the tokenization workflow in Gogo is given by:

$$
\begin{aligned}
x &= \text{Mel}(w), \\
x^1, x^2, \cdots, x^{n_g} &= \text{Grouping}(x), \\
z^i &= \text{Cat}(x^i, q^i), \\
\bar{q}^i, s^i &= \text{FSQ}(\text{Encoder}(z^i)),
\end{aligned}
\tag{1}
$$

where $\text{Mel}(\cdot)$ denotes the mel extraction operation, $\text{Grouping}(\cdot)$ denotes the grouping operation, and $\text{Cat}(\cdot)$ denotes the concatenation operation. We omit the batch dimension for clarity. In practice, the effective batch size for the encoder is $n_g$ times the number of speech samples loaded in a batch.

For reconstruction, $\bar{q}^i$ are first padded with $(g - n_q)$ placeholder tokens to match the original group length $g$, resulting in aligned features $\bar{x}^i \in \mathbb{R}^{g \times d_h}$. All groups' $\bar{x}^i$ are concatenated along the time axis to generate $\bar{x} \in \mathbb{R}^{n_f \times d_h}$, which is then fed into a flow-matching model to predict the mel-spectrogram. The final waveform $\bar{w}$ is recovered using a pretrained Vocos (Siuzdak, 2023) vocoder. Formally, the reconstruction workflow in Gogo is given by:

$$
\begin{aligned}
\bar{x}^i &= \text{Cat}(\bar{q}^i, \text{Placeholders}), \\
\bar{x} &= \text{Cat}(\bar{x}^1, \bar{x}^2, \cdots, \bar{x}^{n_g}), \\
\bar{w} &= \text{Vocoder}(\text{Flow}(\bar{x})).
\end{aligned}
\tag{2}
$$

### 3.1.2 TRAINING OBJECTIVES

Gogo leverages conditional flow matching (CFM) (Lipman et al., 2023), which extends the framework of continuous normalizing flows (Chen et al., 2018) to learn a time-dependent vector field that transports a simple prior distribution, *e.g.*, the standard normal distribution, to the distribution of the mel-spectrograms conditioned on the features $\bar{x}$. Given a mel-spectrogram $x_1$ and a Gaussian noise $x_0 \sim \mathcal{N}(0, I)$, we first interpolate between them and produce a noisy spectrogram $x_t = (1 - t)x_0 + tx_1$ using a flow time $t \sim \mathcal{U}(0, 1)$. The conditional vector field for this linear interpolation is $v(x_0, x_1, t) = \partial x_t / \partial t = x_1 - x_0$. The flow-matching model, parameterized by $\theta$, takes $(x_t, \bar{x}, t)$ as input and predicts a velocity vector $v_\theta(x_t, \bar{x}, t)$. Formally, the CFM objective is defined as a simple vector field regression loss:

$$
\mathcal{L}_{\text{CFM}} = \mathbb{E}_{t, p(x_0), q(x_1)} \left[ \| v_\theta(x_t, \bar{x}, t) - v(x_0, x_1, t) \|_2^2 \right].
\tag{3}
$$

We further introduce two auxiliary modules, AR prior and ASR module, to encourage the speech queries to capture temporal dependencies and linguistic information within each group, respectively.

The corresponding training objectives, $\mathcal{L}_{\mathrm{AR}}$ and $\mathcal{L}_{\mathrm{ASR}}$, are formally defined in Appendix B. Finally, the objective for training Gogo can be written as:

$$\mathcal{L}_{\mathrm{Gogo}} = \lambda_{\mathrm{CFM}}\mathcal{L}_{\mathrm{CFM}} + \lambda_{\mathrm{AR}}\mathcal{L}_{\mathrm{AR}} + \lambda_{\mathrm{ASR}}\mathcal{L}_{\mathrm{ASR}}, \qquad (4)$$

where $\lambda_{\mathrm{CFM}}$, $\lambda_{\mathrm{AR}}$, and $\lambda_{\mathrm{ASR}}$ are coefficients employed to balance different loss components.

### 3.1.3 GRANULARITY ORDERING

To enforce a coarse-to-fine ordering in the learned speech queries $q^i$ and tokens $s^i$, we introduce two techniques: nested dropout (Rippel et al., 2014) and loss balancer.

**Nested dropout** randomly drops tokens in a nested fashion during training. Specifically, we uniformly sample the number of tokens to retain, $n_k \in \{1, \ldots, n_q\}$, and replace the last $(n_q - n_k)$ tokens with masking tokens $m$. This mechanism drives Gogo to prioritize encoding high-level abstractions and essential structural attributes into the earlier coarse tokens to minimize $\mathcal{L}_{\mathrm{Gogo}}$ to the greatest extent, while deferring the challenging and fluctuating details to the later fine tokens. Since later tokens are rarely preserved and receive fewer gradient updates, we further introduce a re-weighting mechanism for compensation. Concretely, the gradient of the $j$-th speech token is scaled by $w_j = 0.5/(1-(j-1)/n_q)$, assigning larger weights to tokens with fewer updates and vice versa. Details of the re-weighting implementation are provided in Appendix A.

**Loss balancer** is utilized to adjust the loss coefficients $\lambda_{\mathrm{CFM}}$ and $\lambda_{\mathrm{ASR}}$, further ensuring that the learned speech tokens are organized in a coarse-to-fine manner. Specifically, when $n_k$ is small, the model should emphasize $\mathcal{L}_{\mathrm{ASR}}$ so that coarse tokens encode richer linguistic content. Conversely, when $n_k$ is large, $\mathcal{L}_{\mathrm{CFM}}$ should dominate to ensure that fine tokens capture more acoustic details. Let $\lambda_{\max}$ and $\lambda_{\min}$ denote the maximum and minimum weighting coefficients, respectively. The loss balancer adaptively adjusts $\lambda_{\mathrm{CFM}}$ and $\lambda_{\mathrm{ASR}}$ as follows:

$$\lambda_{\mathrm{CFM}} = \lambda_{\min} + \frac{(n_k - 1)(\lambda_{\max} - \lambda_{\min})}{n_q - 1}, \quad \lambda_{\mathrm{ASR}} = \lambda_{\max} - \frac{(n_k - 1)(\lambda_{\max} - \lambda_{\min})}{n_q - 1}. \qquad (5)$$

### 3.2 GOGOSPEECH

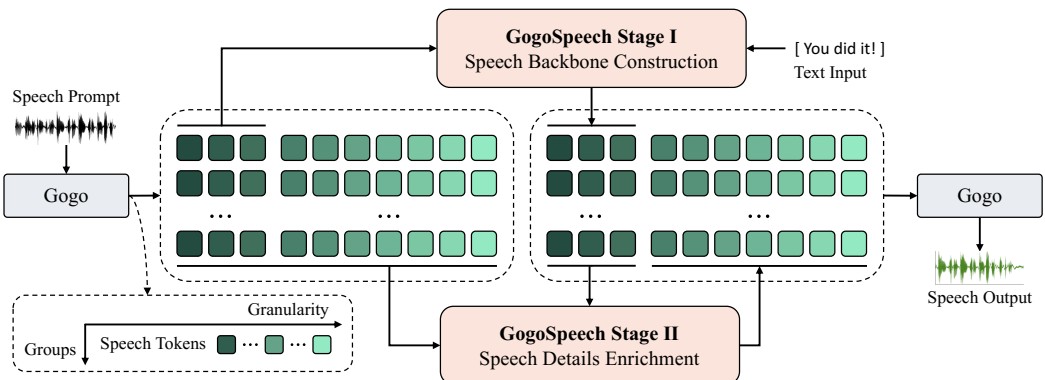

Figure 3: Architecture of GogoSpeech. Gogo encodes the speech prompt into speech tokens, which serve as input to GogoSpeech for generating the target speech tokens. The target speech tokens are transformed into waveform by Gogo. For visualization, the number of speech queries $n_q$ in each group is set to 10 and the speech backbone is defined as the first 3 speech tokens of each group.

### 3.2.1 STAGE I: SPEECH BACKBONE CONSTRUCTION

Stacking all $s^i$ yields a 2D token matrix $\mathbf{S} \in \mathbb{R}^{n_g \times n_q}$. The speech backbone is defined as the first $b$ tokens of each group, *i.e.*, $\mathbf{S}_{:,1:b} \in \mathbb{R}^{n_g \times b}$, which contains the high-level cues of the speech signal. As shown in Figure 3, given the input text $y = (y_1, \ldots, y_L)$ and the backbone of speech prompt $\mathbf{S}_{:,1:b}$, the autoregressive model in Stage I generates the backbone of target speech $\tilde{\mathbf{S}}_{:,1:b} \in \mathbb{R}^{\tilde{n}_g \times b}$

group by group, where $\tilde{n}_g$ denotes the number of groups in the target speech. Let $\Gamma(\cdot)$ denote the operation that flattens a token matrix into a sequence. The objective of Stage I is to minimize the negative log-likelihood over the target speech backbone:

$$\mathcal{L}_{\text{stage1}} = -\sum_{i=1}^{\tilde{n}_g} \sum_{t=1}^{b} \log P(\tilde{\mathbf{S}}_{i,t} \mid y, \Gamma(\mathbf{S}_{:,1:b}), \Gamma(\tilde{\mathbf{S}}_{1:i-1,1:b}), \tilde{\mathbf{S}}_{i,1:t-1}) \tag{6}$$

### 3.2.2 STAGE II: SPEECH DETAILS ENRICHMENT

In Stage II, GogoSpeech progressively enriches the speech backbone predicted in Stage I by adding fine-grained acoustic details group by group. For the $i$-th group of the target speech, the autoregressive model in Stage II generates the fine tokens $\tilde{\mathbf{S}}_{i,b+1:n_q}$ conditioned on all tokens of the input speech prompt $\mathbf{S}$, all tokens of the previously generated groups $\tilde{\mathbf{S}}_{1:i-1,:}$, and the speech backbone of the current group $\tilde{\mathbf{S}}_{i,1:b}$. The training objective for Stage II is given by:

$$\mathcal{L}_{\text{stage2}} = -\sum_{i=1}^{\tilde{n}_g} \sum_{t=b+1}^{n_q} \log P(\tilde{\mathbf{S}}_{i,t} \mid \Gamma(\mathbf{S}), \Gamma(\tilde{\mathbf{S}}_{1:i-1,:}), \tilde{\mathbf{S}}_{i,1:t-1}) \tag{7}$$

### 3.3 TOKEN ALLOCATOR

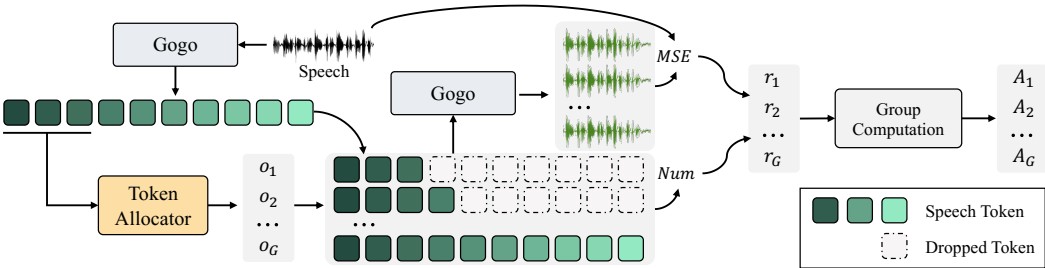

Figure 4: GRPO-trained token allocator. Throughout the GRPO training, the Gogo is kept frozen.

To improve the efficiency of speech generation, we want to allocate more tokens to acoustically complex groups while assigning fewer tokens to simpler ones, such as silence. To enable such adaptive allocation, we design a token allocator that receives the backbone of each group, $\tilde{\mathbf{S}}_{i,1:b}$, as input and outputs a budget $\xi_i \in \{0, 1, \ldots, n_q - b\}$, indicating the number of fine tokens to be generated for that group in GogoSpeech Stage II. The skipped unnecessary fine tokens are replaced with masking tokens $m$. The cooperation between the token allocator $\pi_\omega$ and GogoSpeech is formulated as:

$$\xi_i = \pi_\omega(\tilde{\mathbf{S}}_{i,1:b}) \in \{0, 1, \ldots, n_q - b\}, \tag{8}$$

$$\tilde{\mathbf{S}}_{i,b+1:b+\xi_i} = \arg\max_u \prod_{j=b+1}^{b+\xi_i} p\left(u_j \mid y, \Gamma(\mathbf{S}), \Gamma(\tilde{\mathbf{S}}_{1:i-1,:}), \tilde{\mathbf{S}}_{i,1:j-1}\right), \ \tilde{\mathbf{S}}_{i,b+\xi_i+1:n_q} = m. \tag{9}$$

As shown in Figure 4, the token allocator is trained from scratch using a slightly modified GRPO technique (Shao et al., 2024). Given that the output space of the allocator is relatively small, comprising $(n_q - b + 1)$ discrete allocation choices, we enumerate all possible outputs $(o_1, o_2, \cdots, o_G)$, corresponding to using from $b$ to $n_q$ tokens for reconstructing the input speech. The resulting $(n_q - b + 1)$ reconstructed samples are employed to compute the group scores.

We adopt two reward metrics, including $\mathcal{R}_n$ which penalizes the number of tokens utilized for reconstruction, and $\mathcal{R}_d$ which penalizes the distance between the input and reconstructed speech. This joint reward encourage the allocator to learn allocation strategies that achieve high reconstruction fidelity while minimizing the number of tokens consumed. The rewards $\mathcal{R}_n$ and $\mathcal{R}_d$, and the entire reward $\mathcal{R}$ are defined as follows:

$$\mathcal{R}_n = -\text{Num}(\bar{x}), \ \ \mathcal{R}_d = -\mathbb{E}\left[\|\text{Mel}(w) - \text{Mel}(\bar{w})\|_2^2\right], \tag{10}$$

$$\mathcal{R} = \lambda_n \mathcal{R}_n + \lambda_d \mathcal{R}_d, \tag{11}$$

where $\mathrm{Num}(\bar{x})$ denotes the number of speech tokens used to generate $\bar{w}$, $\lambda_n$ and $\lambda_d$ are coefficients used to balance the two reward terms. After obtaining the reward for each allocation choice, the advantage is calculated using group relative advantage estimation (Shao et al., 2024):

$$\mathcal{A}_j = \frac{\mathcal{R}_j - \mathrm{mean}(\mathcal{R})}{\mathrm{std}(\mathcal{R})}. \tag{12}$$

Since the token allocator is initialized from scratch, we omit the KL penalty term used in the original GRPO framework. The allocator $\pi_\omega$ is then optimized by maximizing the following objective:

$$\mathcal{J}_{\mathrm{GRPO}} = \mathbb{E}_{o_j \sim \pi_\omega(o|\tilde{\mathbf{S}}_{i,1:b})} \left[ \frac{1}{G} \sum_{j=1}^{G} \pi_\omega(o_j|\tilde{\mathbf{S}}_{i,1:b}) \mathcal{A}_j \right]. \tag{13}$$

## 4 EXPERIMENTAL SETUP

### 4.1 IMPLEMENTATION DETAILS

We extract 100-dimensional log mel-filterbank features from audio resampled to 24 kHz, using a hop length of 256 and a window size of 1024, resulting in a feature rate of approximately 94 Hz. We set the group size to $g = 20$, and allocate $n_q = 10$ speech queries per group. Therefore, the token rate of Gogo is computed as $n_q \times (94/g) = 47$ Hz. The speech backbone is defined as the first $b = 3$ tokens of each group. Thus the token rate of backbone is about 14 Hz. Both Stage I and Stage II of GogoSpeech are initialized from the LLaMA (Grattafiori et al., 2024), with the vocabulary expanded to include speech tokens. A pretrained Vocos (Siuzdak, 2023) is employed to convert the generated spectrograms into waveforms. Please refer to Appendix C for more detailed model configuration.

During training, we empirically set the weight of loss $\mathcal{L}_{\mathrm{AR}}$ to $\lambda_{\mathrm{AR}} = 0.06$ and amplify the gradient of the AR prior by a factor of 50 to ensure effective updates. The coefficients for the losses $\mathcal{L}_{\mathrm{ASR}}$ and $\mathcal{L}_{\mathrm{CFM}}$ are dynamically adjusted using the loss balancer defined in Eq. 5, with $\lambda_{\min} = 0.2$ and $\lambda_{\max} = 1.8$. For training the token allocator via GRPO, we set the reward coefficients $\lambda_n = 0.2$ and $\lambda_d = 1.0$ to balance the trade-off between token efficiency and reconstruction quality. Additional hyperparameters, training schedules, and inference configurations are provided in Appendix D.

### 4.2 DATASETS

We train both Gogo, GogoSpeech, and the token allocator on the Emilia dataset (He et al., 2024), a large-scale and diverse in-the-wild speech corpus designed for multilingual speech generation. In this work, we use its English subset, which contains approximately 50K hours of transcribed speech covering a diverse set of speakers, acoustic characteristics, and background conditions. For evaluating the reconstruction quality of Gogo, we adopt the LibriTTS test-clean set (Zen et al., 2019) with 4,837 samples in total. To assess the zero-shot speech generation capability of GogoSpeech, we use the Seed-TTS test-en set (Anastassiou et al., 2024), which consists of 1,000 samples drawn from Common Voice dataset (Ardila et al., 2019). All speech samples are resampled to 24 kHz.

### 4.3 BASELINES AND EVALUATION METRICS

We compare Gogo against multiple codec baselines, including EnCodec (Défossez et al., 2022), DAC (Kumar et al., 2023), SpeechTokenizer (Zhang et al., 2024), Mimi (Défossez et al., 2024), SNAC (Siuzdak et al., 2024), WavTokenizer (Ji et al., 2025), MagiCodec (Song et al., 2025), X-codec2 (Ye et al., 2025b), TAAE (Parker et al., 2025), and DualCodec (Li et al., 2025). All baseline results are obtained using their official checkpoints. Details of compared codecs see Appendix E.

We compare GogoSpeech with TTS baselines, including FireRedTTS-1S (Guo et al., 2025a), F5-TTS Chen et al. (2025), XTTS-v2 (Casanova et al., 2024), Llasa (Ye et al., 2025b), CosyVoice 2 (Du et al., 2024b), and VoiceCraft (Peng et al., 2024). More details can be found in Appendix F.

For evaluation, we employ both objective and subjective metrics. For objective assessment, we adopt UT-MOS (Saeki et al., 2022), DNS-MOS (Reddy et al., 2022), and Perceptual Evaluation of Speech Quality (PESQ) (Rix et al., 2001) to quantify perceptual quality and speech distortion. Speech

Table 1: Comparison between different codec models on the LibriTTS test-clean set. Bold values indicate the best for each token rate. TPS and FPS denote the number of tokens and frames per second, respectively. #CB denotes the number of codebook employed in each model.

| Model | TPS | FPS | #CB | UT MOS | DNS MOS | STOI | PESQ WB | PESQ NB | SIM | WER |
|---|---|---|---|---|---|---|---|---|---|---|
| Ground Truth | - | - | - | 4.13 | 3.83 | 1.00 | 4.64 | 4.55 | 1.00 | 5.86 |
| DAC | 600 | 75 | 8 | **3.78** | **3.75** | **0.99** | **3.52** | **3.85** | **0.98** | **6.10** |
| EnCodec | 600 | 75 | 8 | 3.13 | 3.56 | 0.94 | 2.74 | 3.36 | 0.97 | 6.24 |
| DAC | 150 | 75 | 2 | 1.94 | 3.27 | 0.85 | 1.53 | 1.95 | 0.90 | 10.81 |
| EnCodec | 150 | 75 | 2 | 1.57 | 3.20 | 0.85 | 1.54 | 1.92 | 0.91 | 8.98 |
| SpeechTokenizer | 150 | 50 | 3 | 3.10 | 3.56 | 0.85 | 1.47 | 1.86 | 0.90 | 7.23 |
| Mimi | 150 | 12.5 | 12 | **3.88** | **3.77** | **0.94** | **2.67** | **3.22** | **0.95** | **6.54** |
| SNAC | 82 | 47 | 3 | 3.80 | **3.84** | 0.91 | 2.23 | 2.75 | **0.91** | **7.47** |
| DAC | 75 | 75 | 1 | 1.33 | 2.97 | 0.76 | 1.18 | 1.45 | 0.81 | 30.06 |
| EnCodec | 75 | 75 | 1 | 1.24 | 2.69 | 0.78 | 1.21 | 1.45 | 0.77 | 33.15 |
| WavTokenizer | 75 | 75 | 1 | **4.11** | 3.65 | **0.92** | **2.43** | **2.96** | 0.90 | 8.34 |
| Mimi | 75 | 12.5 | 6 | 3.54 | 3.69 | 0.90 | 2.01 | 2.53 | 0.90 | 7.65 |
| SpeechTokenizer | 50 | 50 | 1 | 1.31 | 3.09 | 0.68 | 1.11 | 1.28 | 0.67 | 9.18 |
| MagiCodec | 50 | 50 | 1 | 4.21 | 3.96 | **0.93** | 2.55 | 3.18 | 0.86 | 7.45 |
| X-codec2 | 50 | 50 | 1 | 4.17 | 3.90 | 0.92 | 2.45 | 3.07 | 0.83 | 6.40 |
| TAAE | 50 | 25 | 2 | **4.27** | 3.89 | 0.91 | 2.14 | 2.82 | 0.87 | 8.18 |
| DualCodec | 50 | 25 | 2 | 4.05 | 3.80 | 0.89 | 2.02 | 2.58 | 0.89 | 6.54 |
| Mimi | 50 | 12.5 | 4 | 3.16 | 3.62 | 0.86 | 1.64 | 2.10 | 0.87 | 9.24 |
| Gogo | 47 | 47 | 1 | 4.19 | **3.99** | 0.92 | **2.59** | **3.26** | **0.91** | 6.35 |

intelligibility is measured using Short-Time Objective Intelligibility (STOI) (Taal et al., 2010) and Word Error Rate (WER). In addition, speaker similarity (SIM) is calculated to evaluate the accuracy of speaker identity preservation. For subjective evaluation, we employ the Similarity Mean Opinion Score (SMOS) and Comparative MOS (CMOS) to assess speaker similarity and relative naturalness, respectively. Detailed definitions for each metric are provided in Appendix G.

## 5 EXPERIMENTAL RESULTS

### 5.1 CODEC COMPARISON

We compare our Gogo with a range of existing codecs, and the results are summarized in Table 1. Despite operating at a relatively low token rate of 47 tokens per second, Gogo achieves superior performance across multiple metrics compared to codecs operating at 50 tokens per second. DAC and EnCodec achieve the best overall performance when operating at a high token rate of 600, with the exception of UT-MOS and DNS-MOS. However, their performance degrades significantly as the token rate is reduced. Notably, Gogo attains UT-MOS and DNS-MOS scores that even surpass the ground-truth recordings, which we attribute to the generative nature of Gogo's flow-matching decoder, enabling enhanced perceptual quality and improved noise robustness.

### 5.2 EFFECTIVENESS OF GROUP-WISE QUANTIZATION FOR AUTOREGRESSIVE MODELING

To evaluate whether group-wise quantization better supports downstream AR modeling, we train a naive autoregressive model to perform AR prediction over speech token sequences, which are constructed by collecting the $j$-th token from each group in Gogo, where $j \in [1, n_q]$. For the frame-wise baseline, we remove the grouping operation in Gogo and replace FSQ with a 10-level RVQ, following the standard frame-wise quantization scheme. Each RVQ layer generates a frame-level token sequence that encodes the residual information left by all preceding layers. Each sequence is modeled autoregressively, and perplexity is reported as an indicator of modeling difficulty. Additional experimental details and perplexity computation are provided in Appendix I.

Table 2: Perplexity of autoregressive modeling on speech tokens produced by different quantization schemes. Column headers specify the source of speech tokens, denoting their positions within each group for group-wise quantization or their corresponding RVQ layer for frame-wise quantization. The boldface denotes the best result. † indicates frame-wise quantization with single-layer VQ.

| Scheme | 1 | 2 | 3 | 4 | 5 | 6 | 7 | 8 | 9 | 10 |
|---|---|---|---|---|---|---|---|---|---|---|
| Frame-wise † | 247.8 | - | - | - | - | - | - | - | - | - |
| Frame-wise | 2.3 | 25.9 | 84.4 | 114.9 | 189.2 | 261.2 | 441.8 | 442.0 | 727.8 | 691.4 |
| Group-wise | **0.9** | **8.5** | **42.0** | **96.9** | **169.5** | **201.6** | **204.1** | **221.9** | **229.6** | **228.3** |

The perplexity results are summarized in Table 2. Group-wise quantization consistently yields lower perplexity across all granularities than frame-wise quantization, suggesting that the group-wise tokens produced by Gogo are more autoregressive-friendly and capture temporal dependencies more effectively. We can also see that coarse tokens yield substantially lower perplexity than fine tokens in both quantization schemes, confirming that fine-grained acoustic details are more challenging for AR models to predict. These findings further motivate the two-stage design of GogoSpeech, where a high-level speech backbone is generated first, followed by fine-grained detail enrichment.

## 5.3 WHAT DO GRANULARITY-ORDERED TOKENS ENCODE?

To gain deeper insight into the behavior of group-wise granularity-ordered quantization, we conduct probing experiments on Gogo's tokens at different granularities to examine the type of information each token encodes. We probe these tokens using a diverse set of acoustic, prosodic, and linguistic features. The probing task is formulated as a regression problem, where the mean squared error reported by the probing model is used as an indicator of each token's representational capacity. More details of the probing setup are provided in Appendix H.

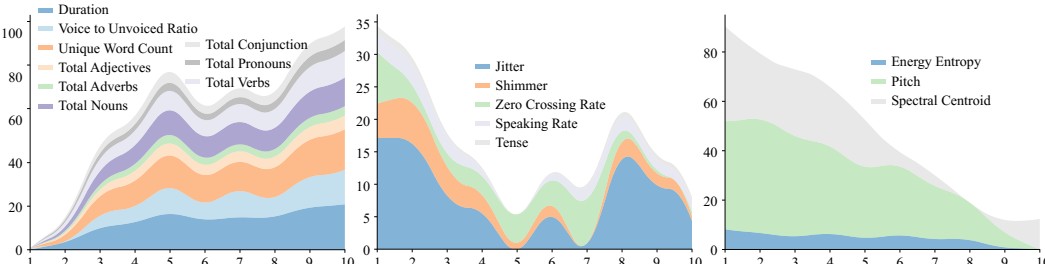

Figure 5: Performance of granularity-ordered tokens across multiple feature prediction tasks. Results are visualized as stacked area charts, where the x-axis denotes token positions within each group and the y-axis indicates the normalized prediction loss relative to the maximum loss for each feature. A higher value corresponds to greater loss and thus lower predictive performance.

The probing results, presented in Figure 5, reveal a clear progression of information across token granularities. Specifically, the first three tokens primarily capture global and high-level information, such as total duration, voiced-to-unvoiced ratio, word count, and linguistic content. Tokens in the middle range predominantly encode prosodic attributes, including speaking rate, jitter, and shimmer. Finally, the last three tokens are responsible for capturing detailed acoustic information, such as pitch, energy, and spectral centroid. These findings confirm that Gogo's group-wise quantization organizes tokens in a coarse-to-fine manner, allowing high-level linguistic and prosodic cues to be modeled with fewer tokens while reserving fine tokens for detailed acoustic reconstruction.

## 5.4 ZERO-SHOT TTS COMPARISON

We compare GogoSpeech with several state-of-the-art TTS baselines, and the results are summarized in Table 3. In objective evaluations, GogoSpeech achieves the highest SIM and competitive WER

Table 3: Comparison between different TTS models on the Seed-TTS test-en set. The boldface denotes the best result, the underline denotes the second best. The Real-Time Factor (RTF) is computed by averaging the inference time of a 47-character sentence over 100 trials on an H100 GPU.

| Model | Objective | | | | | Subjective | |
|---|---|---|---|---|---|---|---|
| | SIM | WER | SIM[†] | WER[†] | RTF | SMOS | CMOS |
| Ground Truth | 0.734 | 2.143 | 0.809 | 2.037 | - | 4.752 | 0.000 |
| F5-TTS (Chen et al., 2025) | 0.647 | **1.830** | 0.716 | 1.812 | **0.184**[‡] | 4.173 | +1.730 |
| XTTS-v2 (Casanova et al., 2024) | 0.463 | 3.248 | 0.490 | 2.292 | 0.208 | 2.426 | −0.961 |
| Llasa-8B-250k (Ye et al., 2025b) | 0.574 | 2.970 | 0.629 | 5.947 | 0.944 | 3.297 | +0.882 |
| CosyVoice 2 (Du et al., 2024b) | 0.654 | 2.380 | 0.701 | 2.324 | 0.549 | 4.331 | +1.638 |
| FireRedTTS-1S (Guo et al., 2025a) | 0.660 | 2.170 | 0.705 | 2.129 | 0.506 | 4.247 | +1.634 |
| VoiceCraft (Peng et al., 2024) | 0.470 | 7.556 | 0.360 | 10.25 | 1.248 | 2.965 | −0.751 |
| GogoSpeech (47 Hz) | **0.667** | 2.394 | **0.725** | **1.788** | 0.535 | **4.381** | **+1.832** |
| w/ Allocator (47 Hz → 36 Hz) | 0.662 | 2.469 | 0.717 | 1.845 | 0.455 | 4.253 | +1.587 |

[†] All target speeches corresponding to the same prompt speech are concatenated, and only constructed samples longer than 10s are retained to evaluate the stability of systems in long speech generation.

[‡] F5-TTS is fully non-autoregressive, while the other systems include autoregressive decoding.

compared to the leading systems. For long-form speech generation, it further attains the best SIM and WER, demonstrating the effectiveness of the two-stage design in enhancing generation stability. In subjective evaluations, GogoSpeech achieves the best SMOS and CMOS scores, confirming its ability to preserve speaker identity while maintaining strong intelligibility and overall quality.

## 5.5 EFFECTIVENESS OF TOKEN ALLOCATOR

We further evaluate the effectiveness of the proposed token allocator by comparing GogoSpeech with and without adaptive token allocation. The results are presented in the last two rows of Table 3. With the token allocator, GogoSpeech generates on average only 36 tokens per second of speech, compared to 47 tokens without adaptive allocation. The token allocator significantly reduces the computational cost of speech generation and incurs only a marginal performance degradation in both objective and subjective scores. As for inference efficiency, F5-TTS is fully non-autoregressive and therefore naturally achieves the lowest RTF. Among the AR systems, GogoSpeech with the token allocator is only slower than XTTS-v2, yet it delivers substantially stronger performance. These results demonstrate that the token allocator achieves a favorable trade-off between generation efficiency and speech quality. Please refer to Appendix L for visualization of adaptive allocation.

## 6 LIMITATIONS

Despite its strong performance, our system has several limitations. First, placeholder tokens in the flow-matching decoder can occasionally introduce artifacts. Second, Gogo operates at a token rate of 47 Hz, which is higher than the low-bitrate codecs of 25 Hz. Finally, GogoSpeech is built on Llama-3.2-1B-Instruct, and its scalability to larger language models requires further investigation.

## 7 CONCLUSION

In this paper, we present Gogo, a group-wise granularity-ordered codec, and GogoSpeech, a two-stage speech language model. Specifically, Gogo produces autoregressive-friendly tokens for each speech group, arranged in order from coarse to fine. Built upon Gogo, GogoSpeech performs speech generation by first constructing a high-level speech backbone, then enriching it with fine-grained details. Furthermore, we proposed a GRPO-trained token allocator that adaptively allocates token budgets based on group-wise complexity, significantly reducing the number of tokens required for synthesis without sacrificing perceptual quality. Extensive experiments on speech reconstruction and zero-shot text-to-speech demonstrate that Gogo achieves superior performance compared to state-of-the-art codecs, and GogoSpeech delivers high-quality, stable, and efficient speech generation.

ETHICS STATEMENT

This work introduces a speech generation model capable of producing highly human-like speech and supporting zero-shot voice cloning. While these capabilities advance the state of the art, they also present potential risks, including misuse for misinformation, impersonation, or other forms of harmful synthetic audio. Our research is intended solely for legitimate scientific purposes.To promote responsible deployment, we advocate transparent disclosure of synthetic speech, appropriate access control, and careful monitoring of downstream use. We are also exploring complementary safeguards such as speech watermarking and deepfake detection to enhance the traceability of generated audio. We encourage the community to adopt similar precautions to ensure that advances in generative speech technology are used ethically and for societal benefit.

This study also involves subjective listening evaluations for assessing the quality of synthesized speech. All participants were fully informed of the purpose and procedure of the listening task, and their participation was entirely voluntary. Consent was obtained prior to the evaluation. Participants were asked to complete the evaluation in a quiet environment to ensure reasonable listening conditions. The study did not collect any personal or identifying information, and no sensitive data or high-risk procedures were involved.

ACKNOWLEDGMENTS

This study was supported in part by both the General Research Fund (Project No. 14202623) from the Research Grants Council and the Centre for Perceptual and Interactive Intelligence, a CUHK-led InnoCentre under the InnoHK initiative of the Innovation and Technology Commission, of the Hong Kong Special Administrative Region Government.

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

## A  NESTED DROUPUOT

Due to the nature of nested dropout, coarse tokens are more likely to be retained during training, whereas fine tokens are more likely to be dropped, leading to fewer effective gradient updates for the latter. To compensate for this imbalance, we rescale the gradient of each token according to its retention probability $p$. Specifically, the coarsest token, *i.e.*, the first learnable token in each group, is always retained, while the finest token, *i.e.*, the last learnable token, is retained with probability $1/n_q$, since we uniformly sample the number of tokens to retain. More generally, the retention probability of the $j$-th token is defined as:

$$p_j = 1 - \frac{j-1}{n_q}, \text{ where } j \in [1, n_q]. \tag{14}$$

We fix the compensation weight for the first token to 0.5, and accordingly define the weight for each token as:

$$w_j = \frac{0.5}{p_j}, \text{ where } j \in [1, n_q]. \tag{15}$$

Let the quantized features of the $i$-th group be $\bar{q}^i \in \mathbb{R}^{n_q \times d_h}$. During the forward pass, we apply the reparameterization trick to scale the token representations as:

$$\bar{q}^i[j] \leftarrow \frac{0.5}{p_j} \bar{q}^i[j] + \left( \bar{q}^i[j] - \frac{0.5}{p_j} \bar{q}^i[j] \right) .\text{detach}, \tag{16}$$

where $\bar{q}^i[j]$ denotes the $j$-th token in group $i$, and the detach operator indicates that the gradient flow is stopped for the detached variable. By applying Equation 16, the value of each embedding $\bar{q}^i[j]$ remains unchanged during the forward pass. However, in the backward pass, its gradient is scaled by the inverse of $p_j$ to offset the imbalance in $p_j$. As a result, $\bar{q}^i[j]$ that receive fewer gradient updates are compensated with a larger gradient scale.

## B  AUXILIARY MODULES

The AR prior takes the quantized representations $\bar{q}^i$ as input and predicts the feature representation of the next speech token at every position. To enhance training stability, the AR prior is optimized with a mean squared error loss in the feature space. Let $f$ denotes the AR prior with parameters $\eta$. The AR loss is defined as:

$$\mathcal{L}_{\text{AR}} = \mathbb{E}_i \left[ \frac{1}{n_q - 1} \sum_{j=1}^{n_q-1} \left\| f_\eta \left( \bar{q}^i[1:j] \right) - \bar{q}^i[j+1] \right\|_2^2 \right]. \tag{17}$$

Furthermore, we incorporate an ASR module into the Gogo training pipeline to facilitate the linguistic representation learning. The group-wise quantized representations $\bar{q}^i \in \mathbb{R}^{n_q \times d_h}$ are concatenated along the temporal dimension to form $\bar{x}_s \in \mathbb{R}^{n_q(n_f/g) \times d_h}$. The ASR module, denoted as $h_\phi$, takes $\bar{x}_s$ as input and outputs a predicted token sequence $\hat{y} = (\hat{y}_1, \ldots, \hat{y}_L)$ corresponding to the ground-truth transcription $y = (y_1, \ldots, y_L)$. The ASR loss is formulated as the cross entropy loss:

$$\mathcal{L}_{\text{ASR}} = \mathbb{E}_{(\bar{x}_s, y)} \left[ -\frac{1}{L} \sum_{t=1}^{L} \log h_\phi(y_t \mid y_{<t}, \bar{x}_s) \right]. \tag{18}$$

## C  MODEL CONFIGURATION DETAILS

For Gogo, the Transformer encoder, ASR module, and AR prior are implemented with 12, 8, and 4 layers of standard Transformer blocks (Vaswani et al., 2017), respectively, following the architecture of LLaMA (Grattafiori et al., 2024). Each layer uses a hidden dimension of 512, 8 attention heads, a feed-forward dimension of 1536, Rotary Position Embeddings (RoPE), RMSNorm, and SwiGLU activation. Asymmetric masking is applied in the Transformer encoder such that mel features can attend to each other but not to the speech queries, whereas each speech query has access to all mel features and to its preceding queries. We set the group size to $g = 20$, and allocate $n_q = 10$ speech queries per group. The feature quantization adopts FSQ with levels $[8, 8, 8, 5, 5]$, yielding an effective codebook size of 12,800. The flow-matching model in Gogo is implemented as a latent Diffusion Transformer (DiT) (Peebles & Xie, 2023), following the configuration of Chen et al. (2025). The only modification is that we set the number of layers to 12. Additionally, the input mel-spectrograms of Gogo are first processed by four ConvNeXt V2 (Woo et al., 2023) layers, followed by a two-layer MLP that projects the features to match the hidden dimension of Gogo.

For GogoSpeech, both Stage I and Stage II are built on top of Llama-3.2-1B-Instruct, with the vocabulary extended to include Gogo's codebook tokens, enabling the model to directly perform speech token generation. GogoSpeech is trained under the next-token prediction paradigm to jointly model text and speech tokens. The maximum sequence length is set to 256 and 1024 tokens for Stage I and Stage II, respectively. The speech backbone is defined as the first $b = 3$ tokens of each group, which are generated in Stage I. The remaining 7 tokens per group are recovered in Stage II.

For the token allocator, we adopt a lightweight Transformer with 2 layers and the same architectural configuration used in Gogo, followed by a linear classifier to predict the token budget for each group.

## D  TRAINING AND INFERENCE DETAILS

Gogo, GogoSpeech, and the token allocator are optimized separately using the AdamW optimizer (Loshchilov & Hutter, 2017a) on 8 NVIDIA H100 NVL 94G GPUs. The learning rate is decayed based on a cosine annealing schedule (Loshchilov & Hutter, 2017b). Speech samples longer than 20 seconds or shorter than 1 seconds are discarded during training. More detailed training hyperparameters are shown in Table 4.

Table 4: Hyperparameters for training Gogo, GogoSpeech, and the token allocator.

| Hyperparameters | Gogo | GogoSpeech Stage I / Stage II | Token Allocator |
|---|---|---|---|
| Training Epochs | - | 10 / 5 | 1 |
| Update Steps | 400k | - | - |
| Warmup Steps | 10k | 5k / 10k | 1k |
| Batch Size | 1440 Seconds | 1152 / 288 Samples | 128 Samples |
| Learning Rate | 2e-4 | 5e-4 | 1e-4 |
| Optimizer | AdamW | AdamW | AdamW |
| Momentum | $\beta_1, \beta_2 = 0.9, 0.999$ | $\beta_1, \beta_2 = 0.9, 0.95$ | $\beta_1, \beta_2 = 0.9, 0.999$ |
| Weight Decay | 0.01 | 0.01 | 0.01 |
| Learning Rate Schedule | Cosine Annealing | Cosine Annealing | Cosine Annealing |

For Gogo inference, we begin from sampled Gaussian noise $x_0$ and integrate it toward the target distribution $x_1$ conditioned on the speech tokens $s^i$. Following Eq. 1 and Eq. 2, the tokens $s^i$ are first converted into aligned representations $\bar{x}$. We then employ an Euler ordinary differential equation (ODE) solver to iteratively integrate $\partial x_t / \partial t = v_\theta(x_t, \bar{x}, t)$ from $x_0$ to $x_1$, where the flow step $t$ is sampled using the Sway Sampling strategy (Chen et al., 2025) to improve generation performance and efficiency. The resulting mel-spectrogram $x_1$ is converted into waveform using a pretrained Vocos vocoder (Siuzdak, 2023). To balance fidelity and diversity, we apply Classifier-Free Guidance (CFG) (Ho & Salimans, 2022) with a guidance scale of 2. For further stability, Exponential Moving Averaged (EMA) weights (Karras et al., 2024) of the model parameters are used during inference.

For GogoSpeech inference, we employ the standard autoregressive decoding strategy widely used in large language models. To balance generation diversity and fidelity, we set the temperature to 0.8, apply a repetition penalty of 1.2 to mitigate degenerate loops, and use nucleus sampling with $p = 1.0$. In Stage II of GogoSpeech, decoding for each group employs early stopping, terminating as soon as the maximum token budget assigned by the token allocator is consumed.

For token allocator inference, the first $b$ tokens of each group (three tokens in our case) produced by Stage I of GogoSpeech are used as input. The allocator then predicts a budget for the group, specifying how many fine tokens should be generated in Stage II.

## E  CODEC BASELINES

**EnCodec** (Défossez et al., 2022) builds upon the residual vector quantization (RVQ) framework and employs a single multi-scale STFT-based discriminator to effectively suppress artifacts and enhance perceptual quality. It supports variable bandwidths by selecting different numbers of codebooks during training, providing a flexible solution for speech compression and discretization in speech language models. For evaluation, we adopt the official implementation and pretrained checkpoint[1].

**DAC** (Kumar et al., 2023) adapts advances from the Improved VQGAN (Yu et al., 2022) image model to address the codebook collapse problem. It performs codebook lookup in a low-dimensional space and replaces Euclidean distance with cosine similarity, improving both stability and quality. DAC further employs a multi-period discriminator in the waveform domain and a multi-band, multi-scale STFT discriminator in the frequency domain, enabling high-fidelity audio generation. For evaluation, we use the official implementation and pretrained checkpoint[2].

**SpeechTokenizer** (Zhang et al., 2024) is specifically designed for speech language modeling, where different aspects of speech are disentangled hierarchically across RVQ layers. Specifically, it employs HuBERT as a semantic teacher to distill content information into the first layer of RVQ. For evaluation, we use the official implementation and pretrained checkpoint[3].

**Mimi** (Défossez et al., 2024) takes inspiration from previous work on SpeechTokenizer and uses distillation to transfer high-level semantic information in WavLM into the quantized tokens. Unlike SpeechTokenizer, it distills semantic information into a plain VQ and apply an RVQ with 7 levels in parallel, thereby removing the constraint that acoustic information must reside in the residual of the semantic quantizer. For evaluation, we use the official implementation and pretrained checkpoint[4].

**SNAC** (Siuzdak et al., 2024) extends RVQ by allowing quantizers to operate at different temporal resolutions. Through a hierarchy of quantizers running at variable frame rates, it adapts to audio structure across multiple timescales, thereby capturing both coarse and fine details more effectively. Each quantizer applies average pooling for downsampling and nearest-neighbor interpolation for upsampling, enabling efficient compression. For evaluation, we adopt the official implementation and pretrained checkpoint[5].

**WavTokenizer** (Ji et al., 2025) improves subjective quality using a single quantizer with an expanded codebook to reduce information loss. It further enhances semantic modeling by introducing attention modules with extended contextual windows in the decoder. Finally, the inverse Fourier transform is employed to reconstruct the final audio directly. For evaluation, we adopt the official implementation and pretrained checkpoint[6].

**MagiCodec** (Song et al., 2025) is a single-layer, streaming Transformer-based audio codec trained with a multistage pipeline to mitigate codebook collapse and improve token efficiency. It introduces Gaussian noise injection and latent regularization techniques to encourage learning low-frequency semantic representations while preventing overfitting to high-frequency noise. For evaluation, we adopt the official implementation and pretrained checkpoint[7].

---

[1] https://github.com/facebookresearch/encodec
[2] https://github.com/descriptinc/descript-audio-codec
[3] https://github.com/ZhangXInFD/SpeechTokenizer
[4] https://huggingface.co/kyutai/mimi
[5] https://github.com/hubertsiuzdak/snac
[6] https://github.com/jishengpeng/WavTokenizer
[7] https://github.com/Ereboas/MagiCodec

**X-codec2** (Ye et al., 2025b) integrates semantic and acoustic features into a unified codebook using a single-layer FSQ quantizer. A pretrained w2v-BERT serves as the semantic encoder, while an acoustic encoder based on residual convolutional blocks with Snake activations captures fine-grained acoustic details. The semantic and acoustic features are concatenated and served as input to the vector quantizer. For evaluation, we adopt the official implementation and pretrained checkpoint[8].

**TAAE** (Parker et al., 2025) introduces a Transformer-based codec architecture that scales into the 1B parameter range, enabling state-of-the-art speech quality at extremely low bitrates. Unlike CNN-based codecs that rely on convolutional inductive biases with high parameter efficiency, TAAE leverages a more general Transformer architecture for greater scalability and better modeling capacity. For evaluation, we adopt the official implementation and pretrained checkpoint[9].

**DualCodec** (Li et al., 2025) is a dual-stream codec that jointly models self-supervised and waveform representations within an end-to-end framework. The first RVQ layer directly encodes semantic-rich features from a pretrained w2v-BERT-2 model, while the remaining RVQ layers, along with the encoder–decoder design, follow the DAC framework. This integration enables DualCodec to better preserve linguistic content while maintaining high-fidelity reconstruction. For evaluation, we use the official implementation and pretrained checkpoint[10].

## F  TTS BASELINES

**FireRedTTS-1S** (Guo et al., 2025a) is a high-quality streamable TTS system that achieves real-time speech generation with low latency under 150ms through text-to-semantic decoding and semantic-to-acoustic decoding. For evaluation, we use the pretrained checkpoint[11].

**F5-TTS** Chen et al. (2025) is a non-autoregressive TTS model built on flow matching. Instead of relying on complex alignment mechanisms, F5-TTS pads the text input with filler tokens to match the length of the target speech and directly performs denoising to generate speech. For evaluation, we use the pretrained checkpoint[12].

**XTTS-v2** (Casanova et al., 2024) is a multilingual zero-shot multi-speaker TTS model built upon the Tortoise model (Betker, 2023). XTTS-v2 supports 16 languages and achieves state-of-the-art results in most of them. For evaluation, we use the pretrained checkpoint[13].

**Llasa** (Ye et al., 2025b) is a large-scale speech synthesis system that employs a single-layer vector quantizer codec and a unified Transformer architecture to fully align with standard LLMs such as Llama. For evaluation, we use the pretrained checkpoint[14].

**CosyVoice 2** (Du et al., 2024b) is a multilingual speech synthesis framework that integrates a pretrained language model for discrete speech token prediction with a chunk-aware flow-matching model for speech feature generation. For evaluation, we use the pretrained checkpoint[15].

**VoiceCraft** (Peng et al., 2024) is a token infilling neural codec language mode. It employs a token rearrangement strategy with causal masking and delayed stacking, enabling seamless speech editing and zero-shot text-to-speech generation. For evaluation, we use the pretrained checkpoint[16].

## G  EVALUATION METRICS

We employ a comprehensive set of evaluation metrics to assess speech quality across multiple dimensions, including intelligibility, perceptual quality, content preservation, and speaker similarity.

---

[8] https://huggingface.co/HKUSTAudio/xcodec2
[9] https://github.com/Stability-AI/stable-codec
[10] https://github.com/jiaqili3/DualCodec
[11] https://github.com/FireRedTeam/FireRedTTS
[12] https://github.com/SWivid/F5-TTS
[13] https://huggingface.co/coqui/XTTS-v2
[14] https://huggingface.co/HKUSTAudio/Llasa-8B
[15] https://huggingface.co/FunAudioLLM/CosyVoice2-0.5B
[16] https://huggingface.co/pyp1/VoiceCraft/blob/main/830M_TTSEnhanced.pth

**Short-Time Objective Intelligibility (STOI)** (Taal et al., 2010) is a widely adopted metric for evaluating speech intelligibility. It computes the correlation between temporal envelopes of reference and reconstructed signals in short-time segments. The score ranges from 0 to 1, with higher values indicating better intelligibility.

**Perceptual Evaluation of Speech Quality (PESQ)** (Rix et al., 2001) measures perceptual speech quality by comparing the reconstructed audio with the clean reference signal using a perceptual auditory model. We report results under both narrow-band (NB, 8 kHz) and wide-band (WB, 16 kHz) conditions.

**UTokyo-SaruLab MOS (UT-MOS)** (Saeki et al., 2022) is an automatic MOS predictor trained to approximate human judgments of overall speech naturalness and quality. It provides a scalable alternative to subjective MOS tests.

**Deep Noise Suppression MOS (DNS-MOS)** (Reddy et al., 2022) is a non-intrusive quality metric designed for real-world audio evaluation. It estimates perceptual quality directly from the signal without requiring reference audio and has been shown to correlate strongly with human ratings.

**Word Error Rate (WER)** is used to quantify content preservation and intelligibility at the linguistic level. By default, we adopt HuBERT as the ASR model[17] to transcribe the reconstructed speech, and compute WER by comparing against the ground-truth transcripts. For zero-shot TTS evaluation on the Seed-TTS test-en set, the WER metric is computed using the provided script[18].

**Speaker Similarity (SIM)** measures the degree to which the synthesized speech retains the identity of the original speaker. By default, we follow Wang et al. (2023) and employ a WavLM-Large-based speaker verification model[19] to extract speaker embeddings from both reconstructed and reference speech, and compute cosine similarity as the final metric. For zero-shot TTS evaluation on the Seed-TTS test-en set, the SIM metric is computed using the provided script[18].

**Similarity Mean Opinion Score (SMOS)** evaluates the speaker similarity between the prompt and the generated speech. Human raters judge the degree of resemblance by considering speaker characteristics, style, acoustic properties, and potential background artifacts. SMOS is scored on a five-point scale, with higher values indicating stronger similarity.

**Comparative Mean Opinion Score (CMOS)** measures the relative perceptual quality of a synthesized sample compared to a reference. Raters assign scores on a scale from -3 to 3, where negative values indicate the synthesized speech is worse than the reference, positive values indicate it is better, and 0 denotes parity.

For subjective evaluation, we randomly select 20 samples from the Seed-TTS test-en set and invite 20 listeners to rate the SMOS and CMOS scores for the synthesized samples.

## H  PROBING EXPERIMENTS

The probing model consists of three fully connected layers with ReLU activations and dropout in between. The hidden dimensions of each layer are 512, 128, and 1, respectively. The probing procedure is formulated as a regression task. Given a speech input, we first extract the target feature to be probed. Next, Gogo is used to quantize the speech and generate multiple groups of tokens. To probe the tokens at position 8, as illustrated in Figure 6, we average the tokens at this position across all groups. The resulting averaged representation is then fed into the probing model to predict the value of the target feature. The mean squared error (MSE) loss of the probing model is reported as an indicator of the representational capacity of tokens at each position, with lower loss implying that the token more readily encodes the probed feature.

We conduct probing experiments on Gogo's granularity-ordered tokens using a broad set of features spanning acoustic, prosodic, and linguistic dimensions. Acoustic features include zero-crossing rate, mean pitch, energy entropy, and spectral centroid, which characterize low-level signal properties. Prosodic features include jitter, shimmer, duration, voiced–unvoiced ratio, and speaking rate,

---

[17]https://huggingface.co/facebook/hubert-large-ls960-ft
[18]https://github.com/BytedanceSpeech/seed-tts-eval
[19]https://github.com/microsoft/UniSpeech/tree/main/downstreams/speaker_verification

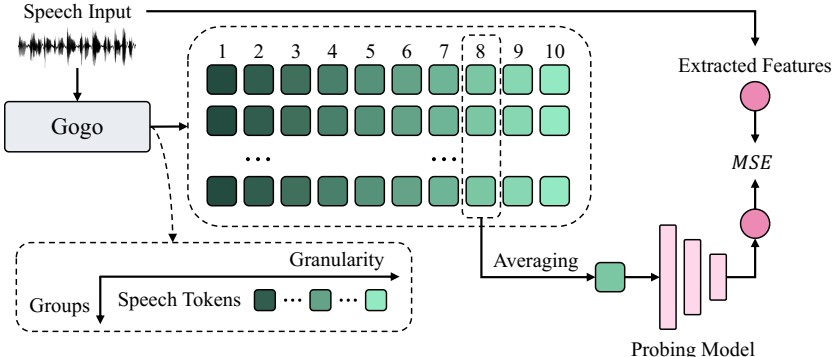

Figure 6: Procedure for probing the information encoded in tokens of different granularities. Throughout the probing model training, the Gogo codec is kept frozen. Here we probe the tokens at position 8 for demonstration.

capturing rhythm, intonation, and phonation stability. Linguistic features include lexical statistics such as unique word count and counts of adjectives, adverbs, nouns, verbs, pronouns, and conjunctions, reflecting higher-level syntactic content. The probing models are trained on the LibriTTS train-clean-100 subset for 20k steps using the AdamW optimizer and evaluated on the LibriTTS test-clean set. The detailed MSE losses for each feature are reported in Table 5.

Table 5: MSE losses for various probed features using Gogo's granularity-ordered tokens on the LibriTTS test-clean set. In our setting, each frame group is discretized into 10 tokens. Token position 1 corresponds to the coarsest token, whereas position 10 corresponds to the finest; ZRC: zero crossing rate, EE: energy entropy, SC: spectral centroid, J: jitter, S: Shimmer, Ratio: voiced to unvoiced ratio, SR: speaking rate, Count: unique word count, Tadj: total adjectives, Tadv: total adverbs, Tn: total nouns, Tv: total verbs, Tpron: total pronouns, Tconj: total conjunction.

| Features | 1 | 2 | 3 | 4 | 5 | 6 | 7 | 8 | 9 | 10 |
|---|---|---|---|---|---|---|---|---|---|---|
| Duration | **14.44** | 14.95 | 15.87 | 16.28 | 16.81 | 16.45 | 16.59 | 16.65 | 17.23 | 17.45 |
| ZRC (e-3) | 1.113 | 1.058 | 1.051 | 1.060 | 1.076 | 1.071 | 1.101 | 1.047 | **1.036** | 1.038 |
| EE (e-3) | 8.409 | 8.290 | 8.189 | 8.257 | 8.142 | 8.212 | 8.101 | 8.066 | 7.839 | **7.771** |
| SC (e-3) | 1.150 | 1.053 | 1.054 | 1.031 | 0.988 | 0.879 | 0.866 | **0.830** | 0.872 | 0.933 |
| Pitch | 1455 | 1478 | 1424 | 1371 | 1304 | 1295 | 1230 | 1166 | 1074 | **1012** |
| J (e-5) | 4.000 | 3.969 | 3.696 | 3.603 | **3.415** | 3.584 | 3.445 | 3.887 | 3.751 | 3.563 |
| S (e-4) | 3.463 | 3.497 | 3.435 | 3.382 | 3.321 | 3.345 | **3.290** | 3.377 | 3.350 | 3.312 |
| Ratio | 0.298 | **0.297** | 0.313 | 0.318 | 0.332 | 0.319 | 0.332 | 0.323 | 0.338 | 0.343 |
| SR | 0.414 | 0.415 | 0.413 | 0.406 | **0.403** | 0.406 | 0.411 | 0.411 | 0.409 | 0.409 |
| Count | **35.81** | 36.90 | 39.01 | 40.00 | 41.18 | 40.34 | 40.63 | 40.70 | 41.96 | 42.47 |
| Tense | 0.240 | 0.239 | 0.238 | 0.237 | **0.236** | 0.237 | 0.237 | 0.239 | 0.239 | 0.238 |
| Tadj | **2.229** | 2.250 | 2.299 | 2.329 | 2.352 | 2.334 | 2.337 | 2.338 | 2.359 | 2.370 |
| Tadv | **1.474** | 1.486 | 1.507 | 1.521 | 1.532 | 1.523 | 1.521 | 1.526 | 1.537 | 1.538 |
| Tn | **9.716** | 9.907 | 10.32 | 10.55 | 10.81 | 10.67 | 10.71 | 10.66 | 10.88 | 10.99 |
| Tv | **4.708** | 4.804 | 4.969 | 5.023 | 5.128 | 5.057 | 5.097 | 5.116 | 5.240 | 5.282 |
| Tpron | **2.766** | 2.797 | 2.838 | 2.847 | 2.868 | 2.847 | 2.853 | 2.866 | 2.897 | 2.906 |
| Tconj | **0.967** | 0.977 | 0.998 | 1.005 | 1.015 | 1.003 | 1.006 | 1.010 | 1.024 | 1.027 |

We further investigate how reconstruction quality varies when only a subset of the granularity-ordered tokens is used. Specifically, we progressively retain the first $n$ tokens in each group and reconstruct the input speech, where $n \in [1, n_q]$. The results are shown in Figure 7. We observe that the word error rate drops sharply when the first few tokens are included, indicating that most linguistic content is captured by the coarsest tokens in Gogo. However, when more than six tokens per group are retained, the improvement of WER becomes marginal, suggesting that the remaining

fine-grained tokens primarily contribute to perceptual quality and acoustic details rather than linguistic content. In particular, both NB PESQ and WB PESQ scores show a marked improvement once more than four tokens per group are retained. The other objective metrics exhibit a generally monotonic improvement as the number of retained tokens increases.

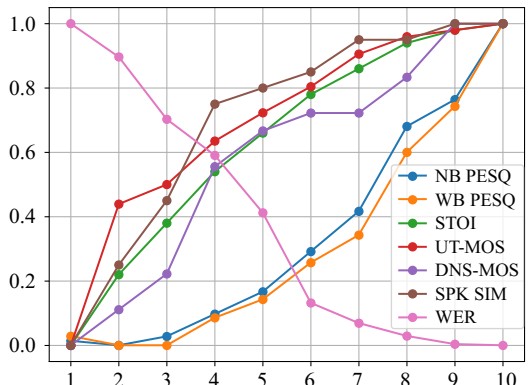

Figure 7: Normalized performance on the LibriTTS test-clean set with varying numbers of retained tokens per group.

## I PERPLEXITY EXPERIMENTS

Perplexity (PPL) is a standard evaluation metric for language models that quantifies their uncertainty in predicting the next token in a sequence. Lower PPL indicates higher model confidence and prediction accuracy, whereas higher PPL reflects greater uncertainty and poorer predictive performance. Formally, perplexity is defined as the exponentiated average negative log-likelihood of the sequence. In our setting, given a speech token sequence $\Gamma(\mathbf{S}_{:,j})$, where $\mathbf{S} \in \mathbb{R}^{n_g \times n_q}$ is the token matrix produced by Gogo and $j \in [1, n_q]$ denotes the position of the speech token within each group, we compute PPL to assess the autoregressive modeling difficulty at different token granularities as follow:

$$\mathrm{PPL}\left(\Gamma(\mathbf{S}_{:,j})\right) = \exp\left\{-\frac{1}{n_g}\sum_i^{n_g} \log P(\mathbf{S}_{i,j}|\mathbf{S}_{<i,j})\right\}. \tag{19}$$

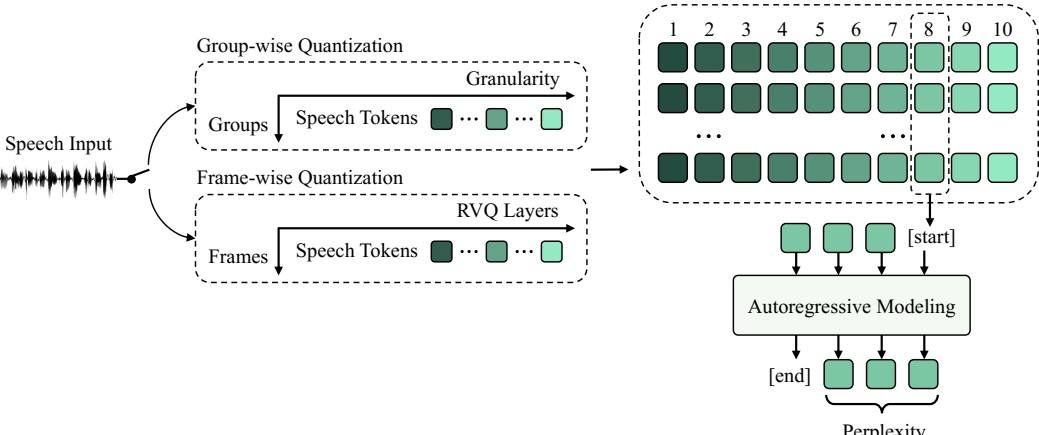

Figure 8: Procedure for evaluating the perplexity of autoregressive model on speech tokens generated by different quantization schemes. Here we show the evaluation using the 8th token of each group or tokens from the 8th RVQ layer for demonstration.

As illustrated in Figure 8, the autoregressive model used in the perplexity experiments is a 6-layer LLaMA-style Transformer with hidden dimension of 256. It is trained on the combined LibriTTS train-clean-100, train-clean-360, and train-other-500 subsets for 200k steps using the AdamW optimizer. Evaluation is conducted on the LibriTTS test-clean set.

## J  STATISTICAL SIGNIFICANCE ANALYSIS

In Table 6, we report the 95% confidence intervals for all subjective evaluation metrics to characterize score variability across listeners. For CMOS, in which listeners provide paired comparative judgments, we evaluate statistical significance using the Wilcoxon signed-rank test and include the corresponding p-values. Across models, several systems display statistically significant differences from the Ground Truth, as indicated by p-values $\leq 0.05$. Our proposed GogoSpeech also achieves a statistically significant positive CMOS relative to the Ground Truth, and incorporating the token allocator leads to a similarly significant improvement.

Table 6: Subjective comparison between different TTS models on the Seed-TTS test-en set. The boldface denotes the best result, the underline denotes the second best. We report 95% confidence intervals for all scores and assess statistical significance (p-value) using the Wilcoxon signed-rank test for CMOS. p-value $\leq 0.05$ indicate statistical significance.

| Model | SMOS | CMOS (p-value) |
|---|---|---|
| Ground Truth | $4.752 \pm 0.342$ | 0.000 |
| F5-TTS (Chen et al., 2025) | $4.173 \pm 0.513$ | $+1.730 \pm 0.476$ (4.9e-4) |
| XTTS-v2 (Casanova et al., 2024) | $2.426 \pm 0.572$ | $-0.961 \pm 1.088$ (0.072) |
| Llasa-8B-250k (Ye et al., 2025b) | $3.297 \pm 0.533$ | $+0.882 \pm 0.869$ (0.027) |
| CosyVoice 2 (Du et al., 2024b) | $\underline{4.331 \pm 0.435}$ | $+1.638 \pm 0.596$ (2.9e-3) |
| FireRedTTS-1S (Guo et al., 2025a) | $4.247 \pm 0.409$ | $+1.634 \pm 0.561$ (3.0e-3) |
| VoiceCraft (Peng et al., 2024) | $2.965 \pm 0.697$ | $-0.751 \pm 0.877$ (0.170) |
| GogoSpeech (47 Hz) | $\mathbf{4.381 \pm 0.479}$ | $\mathbf{+1.832 \pm 0.491}$ **(2.7e-3)** |
| w/ Token Allocator (47 Hz $\rightarrow$ 36 Hz) | $4.253 \pm 0.344$ | $+1.587 \pm 0.530$ (0.013) |

## K  ABLATION EXPERIMENTS

Table 7: Ablation study on the auxiliary modules and granularity ordering. Gogo is evaluated on the LibriTTS test-clean set. GogoSpeech is evaluated on the Seed-TTS test-en set.

| Model | UT MOS | DNS MOS | STOI | PESQ WB | PESQ NB | SIM | WER | SIM | WER |
|---|---|---|---|---|---|---|---|---|---|
| | Gogo | | | | | | | GogoSpeech | |
| | | | | | Auxiliary Module | | | | |
| Proposed Model | **4.19** | **3.99** | **0.92** | 2.59 | 3.26 | 0.91 | **6.35** | **0.667** | **2.394** |
| w/o ASR Module | 3.97 | 3.91 | 0.87 | **2.63** | **3.29** | **0.92** | 8.66 | 0.640 | 4.372 |
| w/o AR Prior | 4.14 | 3.96 | 0.86 | 2.61 | 3.24 | 0.91 | 6.42 | 0.661 | 2.625 |
| | | | | Granularity Ordering | | | | | |
| Proposed Model | **4.19** | **3.99** | 0.92 | **2.59** | **3.26** | **0.91** | **6.35** | **0.667** | **2.394** |
| w/o Nested Dropout | 4.10 | 3.89 | 0.89 | 2.39 | 3.16 | 0.89 | 6.41 | 0.657 | 3.969 |
| w/o Loss Balancer | 3.89 | 3.74 | 0.82 | 2.29 | 3.07 | 0.85 | 7.50 | 0.614 | 4.608 |

**Gogo.** We perform an ablation study on all auxiliary modules and design choices in Gogo and further evaluate their impact on GogoSpeech. The results are summarized in Table 7. We can see that removing the ASR module leads to a slight improvement in PESQ and SIM for Gogo's reconstruction. However, the absence of ASR guidance significantly degrades the performance of GogoSpeech across all metrics. Eliminating the AR prior has little effect on Gogo's reconstruction metrics but

results in a noticeable performance drop for GogoSpeech. This suggests that the AR prior primarily benefits the autoregressive modeling of GogoSpeech rather than signal reconstruction. Furthermore, we ablate the granularity ordering mechanism. We can observe that removing either nested dropout or the loss balancer causes a substantial decline in performance for both Gogo and GogoSpeech, highlighting their importance in learning informative coarse-to-fine token representations.

Table 8: Ablation study on the number of coarse tokens $b$ used as the speech backbone on the Seed-TTS test-en set. Using all 10 tokens as the backbone degenerates into a single-stage model.

| $b$ | 1 | 2 | 3 | 4 | 5 | 6 | 7 | 8 | 9 | 10 |
|-----|-----|-----|-----|-----|-----|-----|-----|-----|-----|-----|
| SIM | 0.647 | 0.654 | **0.667** | 0.663 | 0.662 | 0.652 | 0.657 | 0.656 | 0.654 | 0.642 |
| WER | 2.754 | 2.531 | 2.394 | **2.391** | 2.397 | 2.441 | 2.346 | 2.451 | 2.460 | 3.121 |

**GogoSpeech.** We further perform an ablation study on the number of coarse tokens $b$ used as the speech backbone, with results summarized in Table 8. We observe that increasing $b$ from 1 to 3 yields substantial improvements in both SIM and WER, indicating that a slightly richer backbone provides more effective guidance for Stage II refinement. When $b$ lies between 3 and 5, the performance of both metrics becomes relatively stable, with SIM reaching its peak at $b = 3$ and WER achieving its best value at $b = 4$. As $b$ increases further from 6 to 9, all evaluation metrics exhibit a general downward trend, suggesting that overly detailed backbones may limit the capacity of Stage II to contribute fine-grained refinements. When $b = 10$, meaning that all tokens are generated entirely within Stage I of GogoSpeech, the performance drops to its lowest level across all metrics. These results provide strong empirical evidence for the effectiveness of our two-stage design, where a compact backbone combined with detail refinement yields the best overall performance.

Table 9: Ablation study on the token allocator on the Seed-TTS test-en set. Various token allocators, trained under different experimental settings, are applied to the same GogoSpeech model.

| Model | GogoSpeech | |
|-------|:----------:|:----:|
| | SIM | WER |
| *GRPO Modifications* | | |
| Proposed Model | **0.662** | **2.469** |
|   w/o Removal of KL Penalty | 0.647 | 3.452 |
|   w/o Exhaustive Enumeration | 0.651 | 2.659 |
| *Sensitivity to Reward Weights* | | |
| Proposed Model ($\lambda_n = 0.2, \lambda_d = 1.0$) | **0.662** | **2.469** |
| $\lambda_n = 0.5, \lambda_d = 1.0$ | 0.656 | 2.882 |
| $\lambda_n = 1.0, \lambda_d = 1.0$ | 0.650 | 3.169 |
| $\lambda_n = 1.0, \lambda_d = 0.5$ | 0.642 | 3.772 |
| $\lambda_n = 1.0, \lambda_d = 0.2$ | 0.643 | 3.528 |
| *Reinforcement Learning Objectives* | | |
| Proposed Model | **0.662** | **2.469** |
| Replace GRPO with DPO (Rafailov et al., 2023) | 0.645 | 3.691 |

**Token Allocator.** We further conduct an ablation study to evaluate the effectiveness of our modifications to the standard GRPO algorithm, including the removal of the KL penalty and the use of exhaustive enumeration. We also examine the sensitivity of the token allocator to different reward weights and compare the GRPO-based allocator with its DPO-based counterpart. The results are summarized in Table 9. We observe that removing the KL penalty and introducing exhaustive enumeration during allocator training consistently yield performance gains. Moreover, the reward-sensitivity analysis suggests that greater emphasis should be placed on the reward term $\mathcal{R}_d$, while $\mathcal{R}_n$ should be assigned a smaller weight. Finally, the last row of Table 9 shows that the GRPO-based token allocator outperforms the DPO-based approach.

**Systematic Evaluations under Consistent Training Conditions** To rigorously isolate the impact of our proposed approach and further demonstrate the effectiveness of Gogo, GogoSpeech, and the token allocator, we conduct a carefully controlled ablation study. The results are reported in

Table 10: Systematic evaluation of codec and SLM design choices on the Seed-TTS test-en set. The boldface denotes the best result, the underline denotes the second best.

| Codec | | SLM | | Token | TPS | SIM | WER |
| Frame-wise | Group-wise | Single-stage | Two-stage | Allocator | | | |
| --- | --- | --- | --- | --- | --- | --- | --- |
| ✓ | | ✓ | | | 47 | 0.592 | 4.117 |
| | ✓ | ✓ | | | 47 | 0.642 | 3.121 |
| | ✓ | | ✓ | | 47 | **0.667** | **2.394** |
| | ✓ | | ✓ | ✓ | 36 | 0.662 | 2.469 |

Table 10. All experiments in the table, including codec and SLM model training, are conducted on the Emilia dataset. The frame-wise and group-wise codecs operate at the same token rate, and all SLM models are initialized with Llama-3.2-1B-Instruct. Comparing the first and second rows, both using a single-stage SLM, we observe that the SLM using the group-wise codec (*i.e.*, Gogo) outperforms the one using the frame-wise codec, demonstrating the effectiveness of our group-wise quantization. Comparing the second and third rows, both using the group-wise codec, the two-stage SLM (*i.e.*, GogoSpeech) achieves higher performance metrics than the single-stage SLM, validating the effectiveness of our two-stage design. Finally, comparing the last two rows, the introduction of the token allocator reduces the token rate from 47 Hz to 36 Hz while maintaining roughly the same model performance, indicating the effectiveness of the token allocator.

## L    VISUALIZATION OF TOKEN ALLOCATION

To better illustrate the behavior of the token allocator, we visualize the token allocation results in Figure 9 and Figure 10. Specifically, we present three aligned plots: (1) the original mel-spectrogram of the input speech, where vertical dashed lines indicate group boundaries; (2) the token budget assigned to each group by the allocator; and (3) the reconstructed mel-spectrogram obtained by Gogo using only the allocated tokens. The visualization clearly demonstrates that the allocator adaptively assigns more tokens to acoustically rich regions while reducing the allocation in silent or low-information segments, thereby achieving efficient yet high-quality reconstruction.

## M    THE USE OF LARGE LANGUAGE MODELS

Large language models were employed exclusively as auxiliary tools to edit and polish text written by the authors. Their usage was limited to improving clarity, grammar, and style of expression. No part of the research ideation, methodology, analysis, or results relied on LLMs.

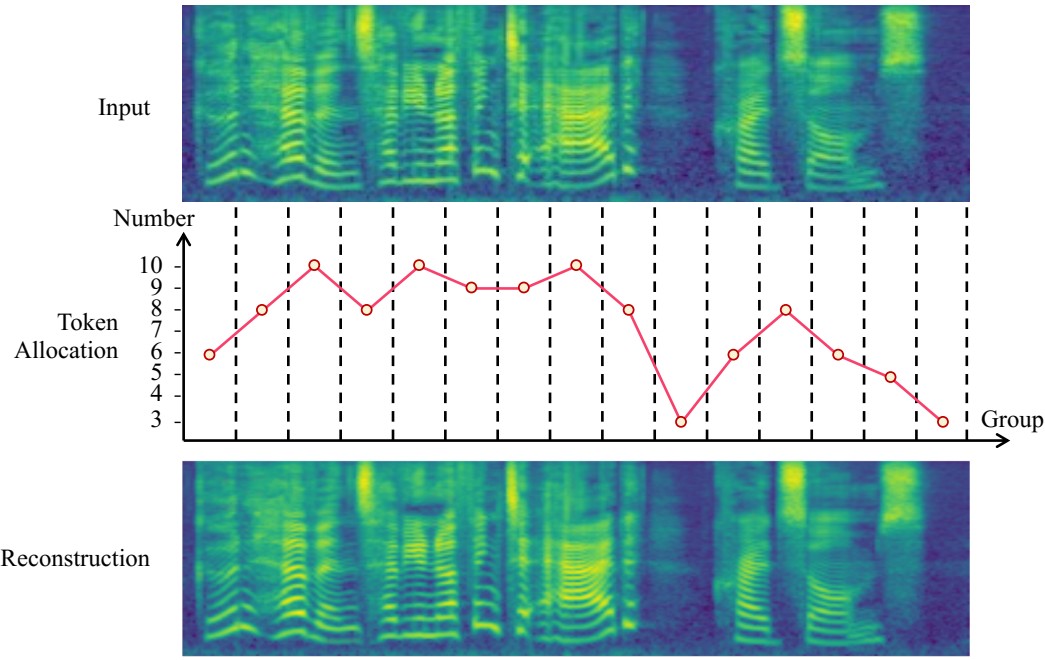

Figure 9: Visualization of sample 1580_141084_000085_000000 from the LibrtTTS test-clean set. The token allocator reduces the token rate from 47 Hz to 34.28 Hz.

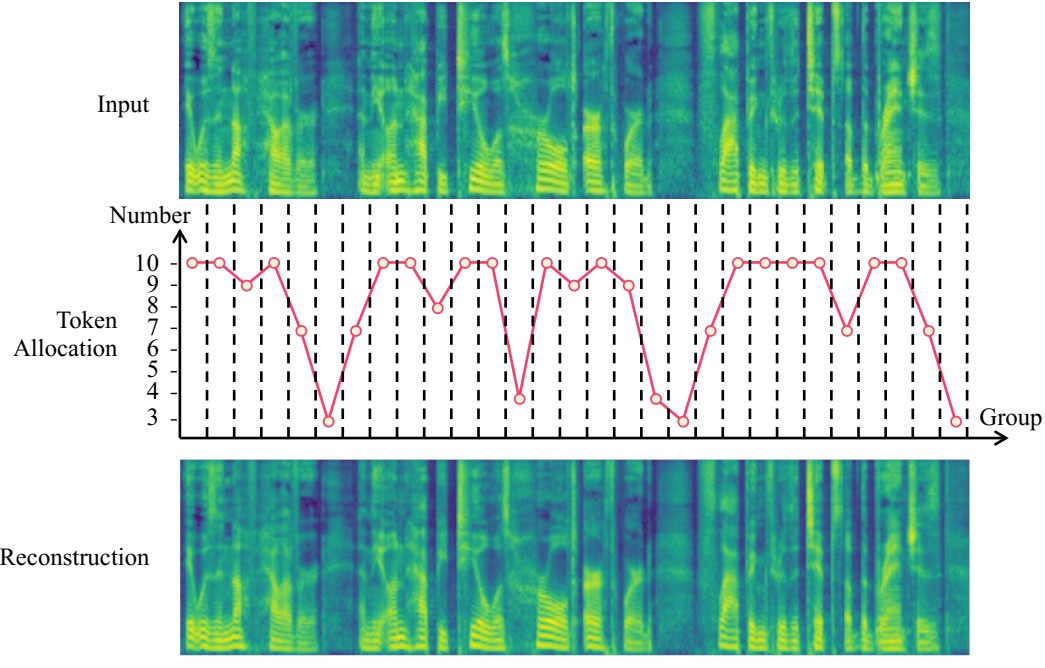

Figure 10: Visualization of sample 7176_88083_000011_000005 from the LibrtTTS test-clean set. The token allocator reduces the token rate from 47 Hz to 38.73 Hz.

