# OpenReview forum: "Gogo: Group-wise granularity-ordered codec for stable and efficient speech generation"
_ICLR.cc/2026/Conference — ICLR 2026 Poster_

### Official Review · Reviewer_tQ8y · 2025-10-21

**Soundness:** 3
**Presentation:** 1
**Contribution:** 2
**Rating:** 4
**Confidence:** 4

**Summary:**

The authors present Gogo, a group-wise granularity-ordered speech codec. The proposed method tokenizes contiguous frames into groups and sequentially orders tokens from coarse to fine granularity. Utilizing Gogo codec, the authors construct GogoSpeech, an LLM-based text-to-speech (TTS) model. GogoSpeech synthesizes speech in two stages: the first stage generates high-level acoustic cues ("speech backbone") at a low token rate, and the second stage refines these cues by adding detailed acoustic information at a standard token rate.

**Strengths:**

Overall, the proposed approach is interesting. GogoSpeech achieves performance comparable to SOTA TTS models.

**Weaknesses:**

(1) Paper Organization\
The main text lacks essential ablation studies and related work sections. The current paper structure does not sufficiently support the primary claims regarding the effectiveness of the proposed approach. I this this paper requires major revision to clearly emphasize core contributions and key experiments in the main body, while relegating supplementary details to the appendix.

(2) Questionable Impact of Asymmetric Masking\
Appendix Table 6 indicates that removing asymmetric masking leads to negligible performance differences (e.g., WER increases from 2.394 to 2.406, indicating no significant difference, described by the authors themselves as "slight"). Given that asymmetric masking is presented as a major component in Section 2.1.3, this result unfortunately undermines its claimed effectiveness. Among the three key techniques introduced in Section 2.1.3, the ineffectiveness of asymmetric masking diminishes the overall contribution.

(3) Insufficient GRPO Details and Experiments\
The authors mention using a "slightly modified GRPO," specifically the removal of the KL penalty and exhaustive enumeration of token budgets. However, critical ablation studies investigating these specific modifications and sensitivity analyses concerning the reward weights are missing. Furthermore, comparisons to alternative reinforcement learning-based objectives (e.g., DOP) are also absent. Such analyses are crucial to substantiate the efficacy and necessity of the proposed GRPO-based token allocator.

(4) Fairness and Depth of Zero-shot TTS Comparison\
In Table 3, GogoSpeech (WER: 2.394) achieves competitive but not consistently superior performance compared to state-of-the-art methods such as F5-TTS (WER: 1.830). Although GogoSpeech slightly outperforms F5-TTS in long-form generation, the overall comparison remains mixed and inconclusive. Due to substantial differences in architecture and training conditions, this heterogeneous comparison does not clearly demonstrate the efficacy of the proposed granularity-ordered tokens.

To rigorously isolate the impact of the proposed approach, systematic experiments varying frame rates, token rates, and codebook sizes under consistent training conditions are required. While claiming state-of-the-art performance would be important, systematic evaluations provide stronger evidence than heterogeneous comparisons. Additionally, if the authors wish to claim only state-of-the-art results, comparisons with API-based TTS systems (e.g., ElevenLabs TTS, OpenAI TTS, Gemini TTS) would significantly enhance the evaluation's rigor and relevance.

**Questions:**

I have no questions but encourage the authors to improve their presentations and experiments.

---

> ### Author Response · Authors · 2025-11-22
> **Official Comment by Authors (1/2)**
>
> **We thank the reviewer for the detailed feedback and constructive questions. We sincerely appreciate the recognition of the strengths of our work.** Below, we provide detailed responses to each of your questions and suggestions.
>
> ---
>
> **Q1. The main text lacks essential ablation studies and related work sections. The current paper structure does not sufficiently support the primary claims regarding the effectiveness of the proposed approach. I think this paper requires major revision to clearly emphasize core contributions and key experiments in the main body, while relegating supplementary details to the appendix.**
>
> **A1.** We sincerely thank the reviewer for the valuable feedback regarding the paper organization. To make the paper more complete and accessible, we have moved the Related Work section from the appendix to the main text. In accordance with your suggestion, we have also relocated certain supplementary details from the main text to the appendix, thereby better highlighting the core contributions of our work (please see our detailed response to Q2).
>
> Regarding the ablation studies, these experiments occupy nearly two pages in length, making it difficult to include them in the main text without disrupting the flow. Therefore, we have placed all ablation studies in the appendix, while ensuring that the main text clearly emphasizes the primary contributions and key experimental results. Thank you again for your precious advice!
>
> ---
>
> **Q2. Appendix Table 6 indicates that removing asymmetric masking leads to negligible performance differences. Among the three key techniques introduced in Section 2.1.3, the ineffectiveness of asymmetric masking diminishes the overall contribution.**
>
> **A2.** We sincerely appreciate your comment. Although asymmetric masking provides some performance improvement, the gain is not sufficiently significant. We have therefore removed the discussion of asymmetric masking from the main text and added the masking configuration used in the Transformer encoder to Appendix C Model Configuration Details. Thank you again for this helpful suggestion, which has helped us better highlight the main contributions of our work and improve the organization of the paper.
>
> ---
>
> **Q3. Insufficient GRPO Details and Experiments. The authors mention using a "slightly modified GRPO," specifically the removal of the KL penalty and exhaustive enumeration of token budgets. However, critical ablation studies investigating these specific modifications and sensitivity analyses concerning the reward weights are missing. Furthermore, comparisons to alternative reinforcement learning-based objectives (e.g., DPO) are also absent. Such analyses are crucial to substantiate the efficacy and necessity of the proposed GRPO-based token allocator.**
>
> **A3.** Thank you very much for your professional comment. We conducted an ablation study to evaluate the effectiveness of our modifications to the standard GRPO algorithm, including the removal of the KL penalty and the use of exhaustive enumeration. We also examined the sensitivity of the token allocator to different reward weights and compared the GRPO-based allocator with its DPO-based counterpart. The results are summarized in the tables below.
>
> **Ablation study on the GRPO modifications:**
> | Model                        | SIM   | WER   |
> |------------------------------|-------|-------|
> | Proposed Model               | **0.662** | **2.469** |
> |   w/o Removal of KL Penalty  | 0.647 | 3.452 |
> |   w/o Exhaustive Enumeration | 0.651 | 2.659 |
>
> **Sensitivity to reward weights:**
> | Model                           | SIM       | WER       |
> |---------------------------------|-----------|-----------|
> | Proposed Model (λ_n=0.2, λ_d=1.0) | **0.662** | **2.469** |
> | λ_n=0.5, λ_d=1.0                  | 0.656     | 2.882     |
> | λ_n=1.0, λ_d=1.0                  | 0.650     | 3.169     |
> | λ_n=1.0, λ_d=0.5                  | 0.642     | 3.772     |
> | λ_n=1.0, λ_d=0.2                  | 0.643     | 3.528     |
>
> **Ablation study on the reinforcement learning objectives:**
> | Model                 | SIM       | WER       |
> |-----------------------|-----------|-----------|
> | Proposed Model        | **0.662** | **2.469** |
> | Replace GRPO with DPO | 0.645     | 3.691     |
>
> We have added these results to Appendix K Ablation Experiments in the updated paper.
>
> ---
>
> Continued in the next comment...

---

> > ### Author Response · Authors · 2025-11-22
> > **Official Comment by Authors (2/2)**
> >
> > **Q4. In Table 3, GogoSpeech (WER: 2.394) achieves competitive but not consistently superior performance compared to state-of-the-art methods such as F5-TTS (WER: 1.830). Although GogoSpeech slightly outperforms F5-TTS in long-form generation, the overall comparison remains mixed and inconclusive. Due to substantial differences in architecture and training conditions, this heterogeneous comparison does not clearly demonstrate the efficacy of the proposed granularity-ordered tokens.**
> >
> > **A4.** Thank you for this thoughtful comment. GogoSpeech is designed with a more stable architecture, particularly for long-form generation where autoregressive and diffusion-based systems often exhibit quality degradation. In long-form settings, GogoSpeech consistently outperforms all compared systems. In addition, in general short-form scenarios, GogoSpeech achieves performance that is comparable to, and in some cases exceeds, current state-of-the-art methods. The results demonstrate that GogoSpeech is a strong and stable alternative.
> >
> > We fully agree that systematic and controlled experiments would provide a more rigorous analysis of the individual factors that influence model performance. However, existing systems differ substantially in their training data, frame rates, token rates, and codebook sizes. Enforcing a unified configuration across all systems would require overriding the settings that their original designs rely on, which could prevent these models from operating under their optimal conditions. Such a setup may unintentionally weaken the baseline systems and thereby reduce the fairness of the comparison.
> >
> > As a complementary effort, we conducted carefully controlled ablation studies to isolate the effects of the proposed components and further demonstrated the effectiveness of Gogo, GogoSpeech, and token allocator.
> > The results are reported in the table below. All experiments in the table, including codec and SLM model training, were conducted on the Emilia dataset. The frame-wise and group-wise codecs operated at the same token rate, and all SLM models were initialized with Llama-3.2-1B-Instruct.
> >
> > |  Frame-wise Codec | Group-wise Codec | Single-stage SLM | Two-stage SLM | Token Allocator | TPS |    SIM    |    WER    |
> > |:-----------------:|:----------------:|:----------------:|:-------------:|:---------------:|:---:|:---------:|:---------:|
> > |         ✓         |                  |         ✓        |               |                 |  47 |   0.592   |   4.117   |
> > |                   |         ✓        |         ✓        |               |                 |  47 |   0.642   |   3.121   |
> > |                   |         ✓        |                  |       ✓       |                 |  47 | **0.667** | **2.394** |
> > |                   |         ✓        |                  |       ✓       |        ✓        |  36 |   0.662   |   2.469   |
> >
> > Comparing the first and second rows, both using a single-stage SLM, we observe that the SLM using the group-wise codec (i.e., Gogo) outperforms the one using the frame-wise codec, demonstrating the effectiveness of our group-wise quantization. Comparing the second and third rows, both using the group-wise codec, the two-stage SLM (i.e., GogoSpeech) achieves higher performance metrics than the single-stage SLM, validating the effectiveness of our two-stage design. Finally, comparing the last two rows, the introduction of the token allocator reduces the token rate from 47 Hz to 36 Hz while maintaining roughly the same model performance, indicating the effectiveness of the token allocator.
> >
> > We have added these results to Appendix K Ablation Experiments in the updated paper.
> >
> > ---
> >
> > **We sincerely thank the reviewer for your time and expertise in providing comments to improve our manuscript. We have tried our best to address all concerns and suggestions to improve our manuscript. Thank you!**

---

> > > ### Author Response · Authors · 2025-11-28
> > > **Follow-Up on Discussion**
> > >
> > > Dear Reviewer tQ8y,
> > >
> > > Thank you once again for your thoughtful and constructive feedback. We have carefully addressed all of your comments in our rebuttal, and the requested clarifications and additional analyses have been incorporated into the updated manuscript as detailed in our responses.
> > >
> > > If you have any further questions or suggestions, we would be very glad to address them before the discussion period concludes. Your insights have been extremely helpful in strengthening our work, and we sincerely appreciate the time and effort you have devoted to reviewing our submission. We look forward to any additional thoughts you may have on the revised version.
> > >
> > > Best regards

---

### Official Review · Reviewer_Mepz · 2025-10-26

**Soundness:** 3
**Presentation:** 3
**Contribution:** 3
**Rating:** 8
**Confidence:** 3

**Summary:**

This paper proposes a new speech codec that groups the mel feature frames into groups and maps them to coarse-to-fine scale tokens. The token sequence then go through the two-stage GogoSpeech model and the outputs are decoded by a flow matching model into mel features after which a Vocos vocoder is used to get the wave file. The codec tokens are learned by some query vectors and the model is trained so that the more frequent and more salient features are expressed in the first few tokens and the granularity increases as the number of tokens representing a speech signal increases. The novelty lies in the design of the codec, and the use of some experimental ideas such as asymmetric masking, nested dropout, a token length based loss balancing terms. The model is trained on the Emilia dataset. Llama is used as the LLM backbone for the autoregressive modeling of the tokens. Experiments compare different codecs with the Gogo codec, where the proposed system achieves competitive performance to other codecs. For codec operating at a similar rate, Gogo slightly outperforms the other ones in terms of PESQ narrowband, speaker similarity, and WER. GogoSpeech TTS model also performs better than the baseline TTS models in terms of subjective MOS scores. On objective metrics (WER and SIM), GogoSpeech either slightly outperforms or is on par with the other TTS models mentioned in the paper.

**Strengths:**

originality,
+ The paper proposes a new speech codec that can represent speech at various granularity.

quality,
+ Experiments demonstrate positive results at a relatively small token rate.

clarity,
+ Mostly clearly written, there are extensive appendices that provide further details.

significance
+ Because of the introduction of a new speech codec, the paper is relevant to the speech community as well as the speech + text joint LLM modeling community as most joint models utilize a form of speech codec.

**Weaknesses:**

- The paper introduces a new codec and experiments positively demonstrate the effectiveness of the codec in audio reconstruction and TTS. As mentioned in the introduction, LLM that accept speech inputs usually make use of a speech codec to tokenize the speech input and hence the paper is relevant to speech LLMs. However, to get a full picture of how this codec would work in a speech + text joint LLM setting (e.g. on a spoken QA task), it could have been better if some further experiments had been included.

- The paper also analyzes what the tokens at different levels learn, which is good. However, it might have been more interesting to see the codec perform in a task where the factors listed in Figure 5 (such as jitter, shimmer, etc.) would more explicitly show its use case. These tasks could have been speech emotion recognition, identification of speech attributes, etc. instead of formulating the probing task as a regression task.

**Questions:**

1. Do the authors think that speech reconstruction and TTS experiments are sufficient to prove that the proposed codec can be successfully utilized in most multimodal LLMs (particularly speech and text LLMs)?

2. Could you please clarify how the regression probing task is set up? Is it based on masking out certain tokens and reconstructing the same audio?

3. In Figure 1, there is a mention of a token budget. Could you please clarify whether once this token allocator is trained, we get the same number of tokens per group? Or is it dynamic within an utterance, i.e., each group getting different number of tokens?

---

> ### Author Response · Authors · 2025-11-22
> **Official Comment by Authors (1/2)**
>
> **We thank the reviewer for the detailed feedback and constructive questions. We sincerely appreciate the recognition of the strengths of our work.** Below, we provide detailed responses to each of your questions and suggestions.
>
> ---
>
> **Q1. Do the authors think that speech reconstruction and TTS experiments are sufficient to prove that the proposed codec can be successfully utilized in most multimodal LLMs (particularly speech and text LLMs)? To get a full picture of how this codec would work in a speech + text joint LLM setting (e.g. on a spoken QA task), it could have been better if some further experiments had been included.**
>
> **A1.** We thank the reviewer for this insightful suggestion regarding spoken QA experiments. We agree that evaluating the codec in a multimodal LLM setting could indeed offer additional perspectives. However, conducting such experiments is beyond the scope of the current study. Spoken QA tasks typically require large-scale, high-quality aligned speech-text datasets and substantial modifications to multimodal LLM architectures, which represent a considerable experimental undertaking.
>
> In addition to our speech reconstruction and TTS evaluations, we also validate the proposed codec through LLM perplexity experiments (Sec. 5.2, Table 2), which show that it is more autoregressive-friendly than existing codecs. Furthermore, we perform probing analyses (Sec. 5.3, Fig. 5) to understand how the codec encodes speech information and why it is effective. We believe these experiments collectively provide strong evidence supporting the codec’s suitability for downstream multimodal LLMs. Extending the evaluation to joint speech-text tasks is an important and promising direction that we plan to explore in future work.
>
> ---
>
> **Q2. Could you please clarify how the regression probing task is set up? Is it based on masking out certain tokens and reconstructing the same audio?**
>
> **A2.** Given a speech input, we first extract the target feature to be probed. Next, Gogo is used to quantize the speech and generate multiple groups of tokens. To probe the tokens at position 8, as illustrated in Appendix Figure 6, we average the tokens at this position across all groups. The resulting averaged representation is then fed into the probing model to predict the value of the target feature. We have added detailed explanation to Appendix H Probing Experiments in the revised manuscript.
>
> ---
>
> **Q3. In Figure 1, there is a mention of a token budget. Could you please clarify whether once this token allocator is trained, we get the same number of tokens per group? Or is it dynamic within an utterance, i.e., each group getting different number of tokens?**
>
> **A3.** In Figure 1, only one group is shown for simplicity, and the token budget corresponds to that single group. In practice, within the same utterance, different groups are assigned different token budgets according to their complexity by the token allocator. We will clarify this explanation in the caption of Figure 1.
>
> ---
> Continued in the next comment...

---

> > ### Author Response · Authors · 2025-11-22
> > **Official Comment by Authors (2/2)**
> >
> > **Q4. The paper also analyzes what the tokens at different levels learn, which is good. However, it might have been more interesting to see the codec perform in a task where the factors listed in Figure 5 (such as jitter, shimmer, etc.) would more explicitly show its use case. These tasks could have been speech emotion recognition, identification of speech attributes, etc. instead of formulating the probing task as a regression task.**
> >
> > **A4.** Thanks for this suggestion. We conduct a probing experiment on speech emotion recognition task. Specifically, we use the Emotional Speech Dataset (ESD), in which each sentence is spoken with five emotional states (neutral, happy, angry, sad, and surprise). ESD contains 350 sentences recorded by 10 English speakers, yielding a total of 350 × 5 × 10 = 17,500 samples. We use the samples from 8 speakers for training and the remaining 2 speakers for testing. The probing model follows the same architecture used in our paper, except that the output layer is modified to predict emotion labels rather than feature values. Correspondingly, we replace the mean squared error loss with cross-entropy loss. The experimental results are presented in the table below.
> >
> > | Token Position | 1     | 2     | 3     | 4     | 5     | 6     | 7     | 8     | 9     | 10    |
> > |----------------|-------|-------|-------|-------|-------|-------|-------|-------|-------|-------|
> > | Accuracy       | 0.264 | 0.287 | 0.244 | 0.304 | 0.315 | 0.334 | 0.338 | 0.339 | 0.355 | 0.352 |
> >
> > Because each sentence in the dataset is spoken with different emotional states, the textual content of the speech sample does not contribute to emotion recognition. Our experimental results align with this observation: the coarse tokens (Position 1-3), which appear in earlier positions and contain high-level linguistic information, exhibit very low recognition accuracy. In contrast, the fine tokens (Position 4-10), which appear in later positions and encode richer acoustic information, consistently achieve higher recognition accuracy.
> >
> > ---
> >
> > **We sincerely thank the reviewer for your time and expertise in providing comments to improve our manuscript. We have tried our best to address all concerns and suggestions to improve our manuscript. Thank you!**

---

> > > ### Comment · Reviewer_Mepz · 2025-11-23
> > >
> > > I would like to thank the authors for responding to the reviews and adding further details based on various questions. I do not have any further comments.

---

### Official Review · Reviewer_UGuC · 2025-10-29

**Soundness:** 3
**Presentation:** 2
**Contribution:** 3
**Rating:** 6
**Confidence:** 5

**Summary:**

The paper introduces Gogo, a new speech tokenizer, and GogoSpeech, a two-stage autoregressive text-to-speech (TTS) framework built on top of it.

The Gogo tokenizer consists of a Transformer encoder that processes grouped mel-spectrogram frames augmented with learnable query tokens, and a flow-based generative model for mel-spectrogram reconstruction. The encoder outputs corresponding to the queries are quantized using Finite Scalar Quantization (FSQ), and combined with filler tokens as conditioning input for the flow model, followed by a pretrained vocoder to synthesize the waveform. To structure token learning, the model enforces a coarse-to-fine hierarchy across the query tokens through asymmetric masking, nested dropout, and adaptive loss balancing, encouraging early tokens to capture global content while later tokens refine acoustic details.

Building on this tokenizer, GogoSpeech employs a two-stage generation process: the first stage predicts a coarse speech backbone from text, while the second stage adds fine-grained acoustic details. Additionally, a reinforcement-learned token allocator dynamically determines the number of fine tokens to generate per segment, improving efficiency by reducing the average token rate from 47 Hz to around 35–36 Hz. Experimental results demonstrate the effectiveness of the proposed framework.

**Strengths:**

I like the idea of using learnable queries to capture the information from grouped mel-spectrogram frames — it’s a neat and flexible way to summarize local acoustic context. Grouping the frames itself is also smart, as it naturally reduces the token rate without much loss of information.

The hierarchical design in the query tokens is particularly interesting. Through asymmetric masking, nested dropout, and adaptive loss weighting, the model explicitly encourages a coarse-to-fine structure across tokens. The probing results in Figure 5 clearly reflect this — earlier tokens capture higher-level linguistic features, while later ones focus more on detailed acoustic attributes like pitch and spectral characteristics.

The addition of the token allocator is another strong point. It’s an elegant way to make the system adaptive by deciding how many fine tokens to generate per segment, which not only saves computation but also reduces the overall token rate without hurting quality.

**Weaknesses:**

1. The paper does not discuss the limitations of the proposed methods, such as potential trade-offs in efficiency, reconstruction artifacts, or scalability.

2. The use of filler or placeholder tokens for the flow-matching decoder may lead to hallucinations or slight pronunciation errors during reconstruction. For example, in the Codec Comparison samples, the first utterance (“he”) sounds closer to “the,” suggesting occasional contextual confusion.

3. The codec architecture involves three major components — a Transformer encoder, FSQ quantizer, and a flow-matching decoder — where the flow-based component requires iterative inference. This could make the overall reconstruction process slower than other non-iterative codecs.

4. GogoSpeech employs two fully autoregressive stages (for backbone and fine-detail prediction), both predicting one token at a time. Since these are cascaded, inference time can be considerably higher than single-stage or non-autoregressive systems.

5. There is no evaluation or comparison of runtime efficiency, inference speed, or computational cost against existing baselines, which makes it difficult to assess the practical performance of the system.

6. Details of the subjective evaluation are missing. The paper does not specify the number of evaluators, number of test samples, listening protocol, or any statistical testing, which limits the reliability of the perceptual results.

7. Statistical significance analysis is not provided for any of the reported metrics. Including confidence intervals or significance tests would help clarify whether the observed improvements are meaningful.

**Questions:**

1. Could the authors clarify how the 4-dimensional grouped input to the Transformer encoder is handled in practice? Specifically, is the `n_g` (group index) dimension flattened into the batch dimension before encoding?

2. Regarding the autoregressive nature of GogoSpeech, the math in Section 2.2 suggests token-level generation within each group. Could the authors confirm whether the model predicts one token at a time across all groups, or all tokens of a single group before moving to the next? It is also unclear how the “flatten” operation affects the sequence length (e.g., whether it becomes *T × b*).

3. The demo or samples page does not load properly — I was only able to access the codec comparison samples, not the others.

**Details Of Ethics Concerns:**

1. Since the paper introduces methods capable of generating highly human-like speech and enabling zero-shot voice cloning, it is important that the authors include an explicit ethics statement addressing potential misuse and responsible deployment of such technology.

2. In addition, because the paper involves subjective listening evaluations, it should include information about fair annotator compensation, consent, and evaluation conditions to ensure transparency and ethical compliance in human subject studies.

---

> ### Author Response · Authors · 2025-11-22
> **Official Comment by Authors  (1/3)**
>
> **We thank the reviewer for the detailed feedback and constructive questions. We sincerely appreciate the recognition of the strengths of our work.** Below, we provide detailed responses to each of your questions and suggestions.
>
> ---
>
> **Q1. Could the authors clarify how the 4-dimensional grouped input to the Transformer encoder is handled in practice? Specifically, is the n_g (group index) dimension flattened into the batch dimension before encoding?**
>
> **A1.** Yes, the n_g dimension is flattened into the batch dimension before being fed into the Transformer encoder. For instance, if B samples are loaded at a time, the effective batch size for the Transformer encoder becomes B×n_g. We have added clear explanation in the revised manuscript.
>
> ---
>
> **Q2. Could the authors confirm whether the model predicts one token at a time across all groups, or all tokens of a single group before moving to the next? It is also unclear how the “flatten” operation affects the sequence length.**
>
> **A2.** At each stage, GogoSpeech must first complete the generation of tokens for the current group before proceeding to generate tokens for the next group. For example, in Stage I, GogoSpeech first generates the first token of group 1, followed by the second and third tokens of group 1, and then proceeds to generate the first token of group 2, and so on. Regarding the “flatten” operation, suppose we have a token matrix S with shape (n_g, n_q), i.e., (group_number, token_number_per_group), as described in section 2.2.1. Flattening S produces a one-dimensional sequence of shape (n_g x n_q). In practice, an additional batch dimension is included, so S has the shape (B, n_g, n_q), and the flattened result has the shape (B, n_g×n_q). We have added detailed explanation in the revised manuscript.
>
> ---
>
> **Q3. The demo or samples page does not load properly. I was only able to access the codec comparison samples, not the others.**
>
> **A3.** Sorry for the inconvenience caused. We found that opening the Anonymous GitHub Page causes all speech samples to be loaded simultaneously, which may result in some samples failing to load. To address this issue, we provide an anonymous link (https://anonymous.4open.science/r/gogo) that allows you to directly access our anonymous repository. You may download the entire repository via this link and play all audio samples locally. Thanks.
>
> ---
>
> **Q4. The codec architecture involves three major components — a Transformer encoder, FSQ quantizer, and a flow-matching decoder — where the flow-based component requires iterative inference. This could make the overall reconstruction process slower than other non-iterative codecs.**
>
> **A4.** Currently, many speech language models (SLMs), such as CosyVoice 2, Kimi-Audio, and GLM-4-Voice, incorporate a flow-matching module at the output stage to enhance speech quality. In our approach, we directly adopt this flow-matching module as the decoder of the codec thereby simplifying the SLM training. For the codec, this design sacrifices reconstruction speed in favor of higher reconstruction quality. For the entire SLM system, the inference time is comparable to that of other systems which also employ flow-matching modules at the output stage. In future work, we also plan to explore more advanced flow-matching techniques, such as one-step flow-matching, to further improve the reconstruction speed.
>
> ---
>
> **Q5. GogoSpeech employs two fully autoregressive stages (for backbone and fine-detail prediction), both predicting one token at a time. Since these are cascaded, inference time can be considerably higher than single-stage systems.**
>
> **A5.** The total number of iterations required to generate speech depends solely on the token rate of the codec and the duration of the target speech. For example, to generate 1 second of speech (assuming it corresponds to 50 tokens), a single-stage autoregressive system must perform 50 iterations. In comparison, GogoSpeech performs 15 iterations in Stage I to generate 15 tokens as the backbone and 35 iterations in Stage II to enrich the backbone, resulting in the same total of 50 iterations/tokens. Therefore, our two-stage design does not introduce a significant increase in inference time.
>
> ---
>
> Continued in the next comment...

---

> > ### Author Response · Authors · 2025-11-22
> > **Official Comment by Authors  (2/3)**
> >
> > **Q6. There is no evaluation or comparison of runtime efficiency, inference speed, or computational cost against existing baselines, which makes it difficult to assess the practical performance of the system.**
> >
> > **A6.** We conducted inference efficiency comparisons. The Real-Time Factor (RTF) is computed by averaging the inference time of a 47-character sentence over 100 trials on an H100 GPU.
> >
> > | Model                   | Type               | RTF   |
> > |-------------------------|--------------------|-------|
> > | F5-TTS                  | Non-autoregressive | 0.184 |
> > | XTTS-v2                 | Autoregressive     | 0.208 |
> > | Llasa-8B-250k           | Autoregressive     | 0.944 |
> > | CosyVoice 2             | Autoregressive     | 0.549 |
> > | FireRedTTS-1S           | Autoregressive     | 0.506 |
> > | VoiceCraft              | Autoregressive     | 1.248 |
> > | GogoSpeech              | Autoregressive     | 0.535 |
> > | GogoSpeech w/ Allocator | Autoregressive     | 0.455 |
> >
> > F5-TTS is fully non-autoregressive and therefore naturally achieves the lowest RTF. Among the autoregressive systems, GogoSpeech with the token allocator is only slower than XTTS-v2, yet it delivers substantially stronger performance across both the SIM and WER metrics.
> >
> > We have added these in the coressponding tables in the updated paper.
> >
> > ---
> >
> > **Q7. Details of the subjective evaluation are missing.**
> >
> > **A7.** Thank you for pointing out the missing details. We randomly selected 20 samples from the Seed-TTS test-en set and invited 20 listeners to conduct the subjective evaluations. We have included these details in the revised manuscript.
> >
> > ---
> >
> > **Q8. Statistical significance analysis is not provided. Including confidence intervals or significance tests would help clarify whether the observed improvements are meaningful.**
> >
> > **A8.** Thank you for this helpful comment. In the revised manuscript, we have added a new section entitled Appendix J Statistical Significance Analysis. In this section, we report 95% confidence intervals for all subjective scores. In addition, for CMOS, in which listeners provide paired comparative judgments, we evaluate statistical significance using the Wilcoxon signed-rank test. The statistical analysis asserts the statistical significance of improvements.
> >
> > ---
> >
> > **Q9. The paper does not discuss the limitations of the proposed methods, such as potential trade-offs in efficiency, reconstruction artifacts, or scalability.**
> >
> > **A9.** Thank you for this comment. In the revised manuscript, we have added a Limitations section at the end of the main text. We also provide the full content of this section below:
> >
> > “Despite its strong performance, our system has several limitations. First, placeholder tokens in the flow-matching decoder can occasionally introduce reconstruction artifacts. Second, Gogo operates at a token rate of 47 Hz, which is higher than typical low-bitrate codecs (12.5 to 25 Hz). Finally, GogoSpeech is built on Llama-3.2-1B-Instruct, and its scalability to larger language models requires further investigation.”
> >
> > ---
> > Continued in the next comment...

---

> > > ### Author Response · Authors · 2025-11-22
> > > **Official Comment by Authors (3/3)**
> > >
> > > **Q10. Since the paper introduces methods capable of generating highly human-like speech and enabling zero-shot voice cloning, it is important that the authors include an explicit ethics statement addressing potential misuse and responsible deployment of such technology. In addition, because the paper involves subjective listening evaluations, it should include information about fair annotator compensation, consent, and evaluation conditions to ensure transparency and ethical compliance in human subject studies.**
> > >
> > > **A10.** Thank you for pointing out the importance of including an ethics statement. In the revised manuscript, we have added an Ethics Statement section at the end of the main text. We also include the full text of the statement below:
> > >
> > > “This work introduces a speech generation model capable of producing highly human-like speech and supporting zero-shot voice cloning. While these capabilities advance the state of the art, they also present potential risks, including misuse for misinformation, impersonation, or other forms of harmful synthetic audio. Our research is intended solely for legitimate scientific purposes.To promote responsible deployment, we advocate transparent disclosure of synthetic speech, appropriate access control, and careful monitoring of downstream use. We are also exploring complementary safeguards such as speech watermarking and deepfake detection to enhance the traceability of generated audio. We encourage the community to adopt similar precautions to ensure that advances in generative speech technology are used ethically and for societal benefit.
> > >
> > > This study also involves subjective listening evaluations for assessing the quality of synthesized speech. All participants were fully informed of the purpose and procedure of the listening task, and their participation was entirely voluntary. Consent was obtained prior to the evaluation. Participants were asked to complete the evaluation in a quiet environment to ensure reasonable listening conditions. The study did not collect any personal or identifying information, and no sensitive data or high-risk procedures were involved.”
> > >
> > > ---
> > >
> > > **We sincerely thank the reviewer for your time and expertise in providing comments to improve our manuscript. We have tried our best to address all concerns and suggestions to improve our manuscript. Thank you!**

---

> > > > ### Comment · Reviewer_UGuC · 2025-11-27
> > > > **Response to Author Comments**
> > > >
> > > > Thank you for the detailed responses and updates to the manuscript. I am satisfied with the clarifications.
> > > >
> > > > I think one of the concerns was missed:
> > > >
> > > > **The use of filler or placeholder tokens for the flow-matching decoder may lead to hallucinations or slight pronunciation errors during reconstruction. For example, in the Codec Comparison samples, the first utterance (“he”) sounds closer to “the,” suggesting occasional contextual confusion.**
> > > >
> > > > Could you provide an explanation for this? And maybe this could be included in the limitations, if this is a persistent problem.

---

> > > > > ### Author Response · Authors · 2025-11-28
> > > > > **Response to Reviewer Comment**
> > > > >
> > > > > Thank you for your insightful comments. The issue you highlighted is indeed a common limitation of flow-matching–based decoders that do not rely on forced alignment. Without explicit local temporal correspondence, the decoder can occasionally produce slight hallucinations or subtle pronunciation shifts.
> > > > >
> > > > > We have included this point in the Limitations section. Thank you again for raising this and helping us improve the manuscript.

---

### Official Review · Reviewer_9Pzb · 2025-11-03

**Soundness:** 2
**Presentation:** 3
**Contribution:** 2
**Rating:** 6
**Confidence:** 4

**Summary:**

This paper proposes Gogo, a novel group-wise granularity-ordered speech codec, and GogoSpeech, a two-stage speech language model for generation. The core idea is to quantize groups of frames into tokens ordered from coarse (high-level information) to fine (acoustic details). GogoSpeech leverages this by first generating a low-rate "speech backbone" (coarse tokens) and then enriching it with fine details in a second stage, aiming to improve stability and efficiency. Additionally, a GRPO-trained token allocator is introduced to adaptively assign token budgets to different speech segments based on their complexity, further enhancing efficiency. Experiments show that Gogo achieves strong reconstruction performance, and GogoSpeech obtains good results in zero-shot TTS.

**Strengths:**

Novelty: The core idea of group-wise, granularity-ordered tokenization is novel and well-motivated. It presents a promising alternative to the dominant frame-wise paradigm.
Systematic Design: The system is thoughtfully designed. The GogoSpeech model is built logically upon the properties of the Gogo codec, and the token allocator further enhances the system's efficiency.

**Weaknesses:**

Modeling Frame Rate: The final token rate of 47 Hz (or 36 Hz with the allocator) is presented as efficient. However, this claim is weakened when compared to recent models that operate at much lower rates, such as 12.5 Hz or even 6.25 Hz. The paper does not adequately position its efficiency gains against these highly competitive low-bitrate models.
Lack of Direct Computational Metrics: The paper relies on token rate as a proxy for efficiency. This is insufficient for a fair comparison. Crucial metrics like Real-Time Factor (RTF) on a standardized hardware setup or computational complexity (e.g., GFLOPS) are missing. Without these, the claim of "efficient generation" remains largely unverified.
Unfair Baseline Comparison in TTS Evaluation: In Table 3, the proposed English-only model is compared against several multilingual TTS systems (e.g., CosyVoice 2). This is not a fair comparison, as multilingual models need to handle the complexities of multiple languages, which can compromise performance on any single language compared to a specialist model. This limitation must be explicitly acknowledged and discussed.

**Questions:**

Could you please provide direct inference efficiency metrics such as RTF (with hardware specifications) or GFLOPS to allow for a more direct and fair comparison of GogoSpeech's computational cost against other models? How does the overall efficiency compare to models with much lower token rates (e.g., 12.5 Hz)?
Regarding the TTS comparison in Table 3, can you comment on the fairness of comparing your English-only system to strong multilingual baselines? Could the comparison be strengthened by evaluating against English-only configurations of these models, if available?
The token allocator is an interesting component. How do you expect it to perform in a multilingual context? Would different speaking habits, prosody, or phonetic structures across languages (e.g., tonal vs. non-tonal) affect its ability to accurately assess group complexity, or would it generalize well without language-specific fine-tuning?

---

> ### Author Response · Authors · 2025-11-22
> **Official Comment by Authors**
>
> **We thank the reviewer for the detailed feedback and constructive questions. We sincerely appreciate the recognition of the strengths of our work.**  Below, we provide detailed responses to each of your questions and suggestions.
>
> ---
>
> **Q1. Could you please provide direct inference efficiency metrics such as RTF (with hardware specifications) or GFLOPS to allow for a more direct and fair comparison of GogoSpeech's computational cost against other models? How does the overall efficiency compare to models with much lower token rates?**
>
> **A1.** Following your suggestion, we conducted inference efficiency comparisons. The RTF is computed by averaging the inference time of a 47-character sentence over 100 trials on an H100 GPU.
>
> | Model                   | Type               | RTF   |
> |-------------------------|--------------------|-------|
> | F5-TTS                  | Non-autoregressive | 0.184 |
> | XTTS-v2                 | Autoregressive     | 0.208 |
> | Llasa-8B-250k           | Autoregressive     | 0.944 |
> | CosyVoice 2             | Autoregressive     | 0.549 |
> | FireRedTTS-1S           | Autoregressive     | 0.506 |
> | VoiceCraft              | Autoregressive     | 1.248 |
> | GogoSpeech              | Autoregressive     | 0.535 |
> | GogoSpeech w/ Allocator | Autoregressive     | 0.455 |
>
> F5-TTS is fully non-autoregressive and therefore naturally achieves the lowest RTF. Among the autoregressive systems, GogoSpeech with the token allocator is only slower than XTTS-v2, yet it delivers substantially stronger performance across both the SIM and WER metrics. In addition, although CosyVoice 2 operates at a lower token rate of 25 Hz compared with our GogoSpeech, its overall inference efficiency remains lower than that of GogoSpeech.
>
> We have added these in the coressponding tables in the updated paper.
>
> ---
>
> **Q2. Regarding the TTS comparison in Table 3, can you comment on the fairness of comparing your English-only system to strong multilingual baselines? Could the comparison be strengthened by evaluating against English-only configurations of these models, if available?**
>
> **A2.** Thank you for pointing this out. We compare GogoSpeech with the current state-of-the-art systems in English TTS. Although some of these systems are multilingual, their English TTS performance is broadly recognized as representing the strongest results achievable to date. We agree that comparing with English-only variants of the multilingual baselines would be more appropriate; however, such versions are not officially provided. Nonetheless, this does not affect the conclusion that GogoSpeech achieves the best reported performance in English TTS.
>
> ---
>
> **Q3. The token allocator is an interesting component. How do you expect it to perform in a multilingual context? Would different speaking habits, prosody, or phonetic structures across languages (e.g., tonal vs. non-tonal) affect its ability to accurately assess group complexity, or would it generalize well without language-specific fine-tuning?**
>
> **A3.** We expect our token allocator to function effectively in multilingual contexts, as both its architecture and its GRPO-based training procedure are inherently language-agnostic. To validate this hypothesis, we conducted the following experiment. We first sampled 1,000 hours of Chinese speech from the Emilia ZH subset to train a Chinese version of the Gogo codec. We then fixed this Chinese Gogo codec and trained a token allocator on the same Chinese training data, denoted as TA-ZH. TA-ZH was subsequently applied to perform token allocation for GogoSpeech-EN in an English TTS setting. In addition, we train a token allocator with the similar setting of 1000 hours of English data, denoted as TA-EN, and compare the performance of GogoSpeech-EN using TA-ZH and that of GogoSpeech-EN using TA-EN. The experimental results are presented below.
>
> | Model                    | SIM   | WER   |
> |--------------------------|-------|-------|
> | GogoSpeech-EN (47 Hz)    | 0.667 | 2.394 |
> | w/ TA-EN (47 Hz → 36 Hz) | 0.662 | 2.469 |
> | w/ TA-ZH (47 Hz → 38 Hz) | 0.665 | 2.424 |
>
> Although the token allocator TA-ZH is trained solely on Chinese data, it is able to reduce the average token rate from 47 Hz to 38 Hz. While this rate is slightly higher than that achieved using TA-EN (36 Hz), the SIM and WER performance of GogoSpeech-EN + TA-ZH is marginally better than that of GogoSpeech-EN + TA-EN. These results suggest that the token allocator generalizes well across languages and can operate effectively in multilingual contexts.
>
> ---
>
> **We sincerely thank the reviewer for your time and expertise in providing comments to improve our manuscript. We have tried our best to address all concerns and suggestions to improve our manuscript. Thank you!**

---

> ### Author Response · Authors · 2025-11-28
> **Follow-Up on Discussion**
>
> Dear Reviewer 9Pzb,
>
> Thank you again for your thoughtful comments, especially regarding efficiency. As highlighted in our rebuttal, we now provide end-to-end RTF measurements. While lowering the token rate can certainly accelerate inference, doing so too aggressively often harms speech quality. This trade-off motivated the design of our token allocator, which performs a dynamic, information-aware balancing between token rate and fidelity.
>
> If you have any further questions or suggestions, we would be happy to address them during the discussion period. Thank you again for your valuable feedback.
>
> Best regards

---

### Public Comment · ~Dongchao_Yang1 · 2025-11-12
**Some discussion about the work**

Hi, I find this is an interesting paper on audio codec design, particularly for introducing a query-based tokenization strategy.

Previously, I conducted a related study titled ALMTokenizer [1], which also converted the traditional frame-to-frame quantization into a query-based compression approach. In your work, this seems closely aligned with your concept (group-wise codec), where additional query tokens are inserted into the original sequence. I’m glad to see that this idea is being further explored and validated in other audio codec research. In the image tokenizer domain, this strategy has already become a standard paradigm due to works such as TiTok [2], and I believe more audio codec research will recognize its advantages over time.

I have two questions and suggestions:

(1) In your setup, you use a relatively large group size (e.g., 20) and insert multiple query tokens (e.g., 10) per group. In ALMTokenizer, we adopted smaller windows (e.g., 5 frames) and inserted one query token per group to maintain flexible frame rates. I’m curious whether these different settings of group size and query count lead to noticeable performance differences?

(2) Although this paper is focused on the speech community, I would suggest adding citations from the multimodal literature, such as TiTok [2] and FlexTok [3]. In particular, if your storyline emphasizes a coarse-to-fine hierarchy with a flow-matching decoder, citing FlexTok would help position your contribution more clearly. Of course, this would not reduce your novelty—the technical realization itself is strong, and from my perspective, this paper has sufficient merit for acceptance at ICLR.


References

[1] Yang D., Liu S., Guo H., et al. ALMTokenizer: A Low-bitrate and Semantic-rich Audio Codec Tokenizer for Audio Language Modeling. ICML 2025.

[2] Yu Q., Weber M., Deng X., et al. An Image Is Worth 32 Tokens for Reconstruction and Generation. NeurIPS 2024.

[3] Bachmann R., Allardice J., Mizrahi D., et al. FlexTok: Resampling Images into 1D Token Sequences of Flexible Length. ICML 2025.

---

> ### Author Response · Authors · 2025-11-28
> **Response to Yang’s Public Comments**
>
> Thank you for your helpful comments. Our work represents a further exploration of query-based quantization and introduces a new paradigm for applying this strategy in the speech domain. In the following, we will respond to your comments one by one.
>
> (1) Increasing both the group size and the number of query tokens indeed yields performance improvements. More importantly, these choices enable additional benefits, such as enabling granularity ordering within each group and decomposing speech generation into two stages (backbone construction followed by detail enrichment). These benefits further improve the overall performance of the system.
>
> (2) We appreciate your suggestion regarding related multimodal literature. The corresponding citations have been added in the revised manuscript.
>
> Thank you again for your attention and constructive feedback.

---

### Meta-Review · Area_Chair_8mAS · 2025-12-15

**Summary:**

The paper proposes Gogo, a group wise, granularity ordered speech codec, and GogoSpeech, a two stage autoregressive TTS framework that leverages coarse to fine hierarchical tokenization and a learned token allocator. Across all reviewers, the work is seen as novel, interesting, and relevant to the speech and speech LLM communities. Reviewers highlight that the approach offers a new perspective on codec design, delivers competitive performance, and demonstrates clear potential for further impact. The reviewers also raise concerns about efficiency claims, experimental rigor, missing or weak ablations, and presentation clarity.  The authors did a good job on the rebuttal and addressed the most of concerns.

**Reviewer Concerns:**

The weak ablations and presentation clarity have been addressed by the rebuttal.

**Reviewer Scores:**

I think the reviewer could have change their scores.

---

### Decision · Program_Chairs · 2026-01-26

Accept (Poster)